# No Free Lunch in LLM Watermarking: Trade-offs in Watermarking Design Choices

Qi Pang    Shengyuan Hu    Wenting Zheng    Virginia Smith
Carnegie Mellon University
{qipang, shengyuanhu, wenting, smithv}@cmu.edu

## Abstract

Advances in generative models have made it possible for AI-generated text, code, and images to mirror human-generated content in many applications. *Watermarking*, a technique that aims to embed information in the output of a model to verify its source, is useful for mitigating the misuse of such AI-generated content. However, we show that common design choices in LLM watermarking schemes make the resulting systems surprisingly susceptible to attack—leading to fundamental trade-offs in robustness, utility, and usability. To navigate these trade-offs, we rigorously study a set of simple yet effective attacks on common watermarking systems, and propose guidelines and defenses for LLM watermarking in practice.

## 1 Introduction

Modern generative modeling systems have notably enhanced the quality of AI-produced content [4, 35, 28, 27]. For example, large language models (LLMs) like those powering Chat-GPT [27] can generate text closely resembling human-crafted sentences. While this has led to exciting new applications of machine learning, there is also growing concern around the potential for misuse of these models, leading to a flurry of recent efforts on developing techniques to detect AI-generated content. A promising approach in this direction is to embed invisible *watermarks* into model-derived content, which can then be extracted and verified using a secret watermark key [16, 9, 5, 19, 45, 17, 12, 41, 38].

In this work, we identify that many of the key properties that make existing LLM watermarks successful can also render them susceptible to attack. In particular, we study a number of simple attacks that take advantage of common design choices of existing watermarking schemes, including:

1. **Robustness** of the watermarks to potential modifications in the output text, so that the watermarks cannot be easily removed [19, 16, 45, 19, 5, 12];
2. The use of **multiple keys** to prevent against watermark stealing attacks [16, 19, 14, 34, 10];
3. **Public detection APIs**, which allow the general public to easily verify whether or not candidate text is AI-generated [16, 36, 24].

While these common features and design choices of existing watermarking schemes have clear benefits, we show that they also make the resulting systems vulnerable to a number of simple but effective attacks. In particular, we study two types of attacks: 1) *watermark-removal attacks*, which remove the watermark from the watermarked content, and 2) *spoofing attacks*, which create (potentially toxic) content with a target watermark embedded, thus making the content appear to be generated by a specific LLM when it is in fact not (see motivating examples in Table 1).

Our work rigorously explores a number of simple removal and spoofing attacks for LLM watermarks. In doing so, we identify critical trade-offs that emerge between watermark robustness, utility, and usability as a result of watermarking design choices. To navigate these trade-offs, we

38th Conference on Neural Information Processing Systems (NeurIPS 2024).

| Prompt | `Alan Turing was born in ...` |
|---|---|
| **Unwatermarked**
Z-Score: 0.16 ↓
PPL: 3.19 | *Alan Turing was born in* 1912 and died in 1954. He was an English mathematician, logician, cryptanalyst, and computer scientist. In 1938, Turing joined the Government Code and Cypher School (GC&CS), where he contributed to the design of the bombe, a machine that was used to decipher the Enigma-enciphered messages... |
| **Watermarked**
Z-Score: 5.98 ↑
PPL: 4.46 | *Alan Turing was born in* 1912 and died in 1954, at the age of 41. He was the brilliant British scientist and mathematician who is largely credited with being the father of modern computer science. He is known for his contributions to mathematical biology and chemistry. He was also one of the pioneers of computer science... |
| **(a) Piggyback spoofing attack**
Exploiting robustness
Z-Score: 5.98 ↑ | *Alan Turing was born in* 1950 and died in 1994, at the age of 43. He was the brilliant American scientist and mathematician who is largely credited with being the father of modern computer science. He is known for his contributions to mathematical biology and musicology. He was also one of the pioneers of computer science... |
| **(b) Watermark-removal attack**
Exploiting multiple keys
Z-Score: 2.40 ↓
PPL: 4.05 | *Alan Turing was born in* 1912 and died in 1954. He was a mathematician, logician, cryptologist and theoretical computer scientist. He is famous for his work on code-breaking and artificial intelligence, and his contribution to the Allied victory in World War II. Turing was born in London. He showed an interest in mathematics... |
| **(c) Watermark-removal attack**
Exploiting public detection API
Z-Score: 1.47 ↓
PPL: 4.57 | *Alan Turing was born in* 1912 and died in 1954. He was an English mathematician, computer scientist, cryptanalyst and philosopher. Turing was a leading mathematician and cryptanalyst. He was one of the key players in cracking the German Enigma Code during World War II. He also came up with the Turing Machine... |

**Table 1:** Examples generated using LLAMA-2-7B with/without the KGW watermark [16] under various attacks. We mark tokens in the green and red lists (see Appendix C). Z-score reflects the detection confidence of the watermark, and perplexity (PPL) measures text quality. (a) In the *piggyback spoofing attack*, we exploit watermark robustness by generating incorrect content that appears as watermarked (matching the z-score of the watermarked baseline), potentially damaging the reputation of the LLM. Incorrect tokens modified by the attacker are marked in orange and watermarked tokens in blue. (b-c) In *watermark-removal attacks*, attackers can effectively lower the z-score below the detection threshold while preserving a high sentence quality (low PPL) by exploiting either the (b) use of multiple keys or (c) publicly available watermark detection API.

propose potential defenses as well as a set of general guidelines to better enhance the security of next-generation LLM watermarking systems. Overall, we make the following contributions:

- We study how watermark *robustness*, despite being a desirable property to mitigate removal attacks, can make the resulting systems highly susceptible to *piggyback spoofing attacks*, a simple type of attack that makes makes watermarked text toxic or inaccurate through small modifications, and show that challenges exist in detecting these attacks given that a single token can render an entire sentence inaccurate (Sec. 4).

- We show that using *multiple watermarking keys* can make the system susceptible to *watermark removal attacks* (Sec. 5). Although a larger number of keys can help defend against watermark stealing attacks, which can be used to launch either spoofing or removal attacks, we show both theoretically and empirically that this in turn increases the potential for watermark removal attacks.

- Finally, we identify that *public watermark detection APIs* can be exploited by attackers to launch both *watermark-removal and spoofing attacks* (Sec. 6). We propose a defense using techniques from differential privacy to effectively counteract spoofing attacks, showing that it is possible to avoid the possibilities of noise reduction by applying pseudorandom noise based on the input.

Throughout, we explore our attacks on three state-of-the-art watermarks [16, 45, 19] and two LLMs (LLAMA-2-7B [37] and OPT-1.3B [44])—demonstrating that these vulnerabilities are common to existing LLM watermarks, and providing caution for the field in deploying current solutions in practice without carefully considering the impact and trade-offs of watermarking design choices. Our code is available at `https://github.com/Qi-Pang/LLM-Watermark-Attacks`.

## 2 Related Work

Advances in large language models (LLMs) have given rise to increasing concerns that such models may be misused for purposes such as spreading misinformation, phishing, and academic cheating.

In response, numerous recent works have proposed watermarking schemes as a tool for detecting LLM-generated text to mitigate potential misuse [16, 9, 5, 19, 45, 17, 12, 41, 38]. These approaches involve embedding invisible watermarks into the model-generated content, which can then be extracted and verified using a secret watermark key. Existing watermarking schemes share a few natural goals: (1) the watermark should be *robust* in that it cannot be easily removed; (2) the watermark should not be easily *stolen*, thus enabling spoofing or removal attacks; and (3) the presence of a watermark should be *easy to detect* when given new candidate text. Unfortunately, we show that existing methods that aim to achieve these goals can in turn enable simple but effective attacks.

**Removal attacks.** Several recent works have highlighted that paraphrasing methods may be used to evade the detection of AI-generated text [18, 13, 20, 21, 43], with [18, 43] demonstrating effective watermark removal using a local LLM. These methods usually require additional training for sentence paraphrasing which can impact sentence quality, or assume a high-quality oracle model to guarantee the output quality is preserved. In contrast, the simple and scalable removal attacks herein do not require additional training or a high-quality oracle. Additionally, our work differs in that we aim to directly connect and study how the inherent properties and design choices of watermarking schemes (such as the use of multiple keys and detection APIs) can inform such removal attacks.

**Spoofing attacks.** Prior works on spoofing use watermark stealing attacks to first estimate the watermark pattern and then embed it into an arbitrary content to launch spoofing attacks. These attacks usually require the attacker to pay a large startup cost by obtaining a significant number of watermarked tokens. For example, [34] requires 1 million queries to the watermarked LLM, and [14, 10] assume the attacker can obtain millions of watermarked tokens to estimate their distribution. Unlike these works, we explore spoofing attacks that are less flexible but can be launched with significantly less upfront cost. In Sec. 4, we explore a very simple and scalable form of spoofing exploiting the inherent robustness property of watermarks, which we refer to as a 'piggyback spoofing attack'. In Sec. 6, we then explore more general spoofing attacks, which instead of querying the watermarked LLM numerous times, consider exploiting the public detection API. In both, our attacks do not require the attacker to estimate the watermark pattern, but share a similar ultimate goal with the prior spoofing attacks to create falsified inaccurate or toxic content that appears to be watermarked.

## 3 Preliminaries

Before exploring attacks and defenses on watermarking systems, we introduce relevant background on LLMs, notation we use throughout the work, and a set of concrete threat models.

**Notation.** We use $\mathbf{x}$ to denote a sequence of tokens, $\mathbf{x}_i \in \mathcal{V}$ is the $i$-th token in the sequence, and $\mathcal{V}$ is the vocabulary. $M_{\mathrm{orig}}$ denotes the original model without a watermark, $M_{\mathrm{wm}}$ is the watermarked model, and $sk \in \mathcal{S}$ is the watermark secret key sampled from the key space $\mathcal{S}$.

**Language Models.** Current state-of-the-art (SOTA) LLMs are auto-regressive models, which predict the next token based on the prior tokens. We define language models more formally below:

**Definition 1** (LM). *We define a language model (LM) without a watermark as:*

$$M_{orig} : \mathcal{V}^* \to \mathcal{V}, \tag{1}$$

*where the input is a sequence of length $t$ tokens $\boldsymbol{x}$. $M_{orig}(\boldsymbol{x})$ first returns the probability distribution for the next token $\boldsymbol{x}_{t+1}$ and then the LM samples $\boldsymbol{x}_{t+1}$ from this distribution.*

**Watermarks for LLMs.** In this work, we focus on three SOTA decoding-based watermarking schemes: KGW [16], Unigram [45] and Exp [19]. Informally, decoding-based watermarks are embedded by perturbing the output distribution of the original LLM. The perturbation is determined by secret watermark keys held by the LLM owner. Formally, we define the watermarking scheme:

**Definition 2** (Watermarked LLMs). *The watermarked LLM takes token sequence $\boldsymbol{x} \in \mathcal{V}^*$ and secret key $sk \in \mathcal{S}$ as input, and outputs a perturbed probability distribution for the next token. The perturbation is determined by $sk$:*

$$M_{wm} : \mathcal{V}^* \times \mathcal{S} \to \mathcal{V} \tag{2}$$

The watermark detection outputs the statistical testing score for the null hypothesis that the input token sequence is independent of the watermark secret key, which reflects the watermark confidence:

$$f_{\mathrm{detection}} : \mathcal{V}^* \times \mathcal{S} \to \mathbb{R} \tag{3}$$

Please refer to Appendix C for additional details of the specific watermarks explored in this work.

### 3.1 Threat Model

**Attacker's Objective.** We study two types of attacks—watermark-removal attacks and (piggyback or general) spoofing attacks. In the watermark-removal attack, the attacker aims to generate a high-quality response from the LLM *without* an embedded watermark. For the spoofing attacks, the goal is to generate a harmful or incorrect output that has the victim organization's watermark embedded. We present concrete application scenarios for attacker's motivations in Appendix B.

**Attacker's Capabilities.** We study attacks by exploiting three common design choices in watermarks: 1) robustness, 2) the use of multiple keys, and 3) public detection APIs. Each attack requires the adversary to have different capabilities, but we make assumptions that are practical and easy to achieve in real-world deployment scenarios.

1) For piggyback spoofing attacks exploiting *robustness* (Sec. 4), we assume that the attacker can make $\mathcal{O}(1)$ queries to the target watermarked LLM. We also assume that the attacker can edit the generated sentence (e.g., insert or substitute tokens).

2) For watermark-removal attacks exploiting *the use of multiple keys* (Sec. 5), we consider the scenario where multiple watermark keys are utilized to embed the watermark, which is a common practice in designing robust cryptographic protocols and is suggested by SOTA watermarks [19, 16] to improve resistance against watermark-stealing attacks [14, 10, 34]. For a sentence of length $l$, we assume that the attacker can make $\mathcal{O}(l)$ queries to the watermarked LLM.

3) For the attacks on *detection APIs* (Sec. 6), we assume that the detection API is available to normal users and the attacker can make $\mathcal{O}(l)$ queries for a sentence of length $l$. The detection returns the watermark confidence score (p-value or z-score). For spoofing attacks exploiting the detection APIs, we assume that the attacker can auto-regressively synthesize (toxic) sentences. For example, they can run a local (small) model to synthesize such sentences. For watermark-removal attacks exploiting the detection APIs, we also assume that the attacker can make $\mathcal{O}(l)$ queries to the watermarked LLM. As is common practice [25, 31] and also enabled by OpenAI's API [26], we assume that the top 5 tokens at each position and their probabilities are returned to the attackers.

## 4 Attacking Robust Watermarks

The goal of developing a watermark that is robust to output perturbations is to defend against watermark removal, which may be used to circumvent detection schemes for applications such as phishing or fake news generation. Robust watermark designs have been the topic of many recent works [45, 16, 19, 34, 17, 32]. We formally define watermark robustness in the following definition.

**Definition 3** (Watermark robustness). *A watermark is $(\epsilon, \delta)$-robust, given a watermarked text $\boldsymbol{x}$, if for all its neighboring texts within the $\epsilon$ editing distance, the probability that the detection fails to detect the edited text is bounded by $\delta$, given the detection confidence threshold $T$:*

$$\forall \boldsymbol{x}, \boldsymbol{x}' \in \mathcal{V}^*, \ \Pr[f_{detection}(\boldsymbol{x}', sk) < T] < \delta, \quad s.t. \ f_{detection}(\boldsymbol{x}, sk) \geq T, \ d(\boldsymbol{x}, \boldsymbol{x}') \leq \epsilon$$

More robust watermarks can better defend against editing attacks, but this seemingly desirable property can also be easily misused by malicious users to launch simple *piggyback spoofing attacks*—e.g., a small portion of toxic or incorrect content can be inserted into the watermarked material, making it seem like it was generated by a specific watermarked LLM. The toxic content will still be detected as watermarked, potentially damaging the reputation of the LLM service provider. As discussed in Sec. 2, spoofing attacks explored in prior work usually require the attacker to obtain millions of watermarked tokens upfront to estimate the watermark pattern [14, 34, 10]. In contrast, our simple piggyback spoofing only requires a single query to the watermarked LLM with careful text modifications, and the effectiveness relates directly to the robustness of the LLM watermark.

**Attack Procedure.** *(i)* The attacker queries the target watermarked LLM to receive a high-entropy watermarked sentence $\mathbf{x}_{wm}$, *(ii)* The attacker edits $\mathbf{x}_{wm}$ and forms a new piece of text $\mathbf{x}'$ and claims that $\mathbf{x}'$ is generated by the target LLM. The editing method can be defined by the attacker. Simple strategies could include inserting toxic tokens into the watermarked sentence $\mathbf{x}_{wm}$ at random positions, or editing specific tokens to make the output inaccurate (see example in Table 1). As we show, editing can also be done at scale by querying another LLM like GPT4 to generate fluent output.

We present the formal analysis on the attack feasibility in Appendix D and point out the takeaway that is universally applicable to all robust watermarks: A more robust watermark makes piggyback

spoofing attack easier by allowing more toxic tokens to be inserted. This is a fundamental design trade-off: If a watermark is robust, such spoofing attacks are inevitable and may be extremely difficult to detect, as even one toxic token can render the entire content harmful or inaccurate.

## 4.1 Evaluation

**Experiment Setup.** We assess the effectiveness of our piggyback spoofing attack by using the two editing strategies discussed above. Through toxic token insertion, we study the limits of how many tokens can be inserted into the watermarked content. Using fluent inaccurate editing, we show that piggyback spoofing can generate fluent, watermarked, but inaccurate results at scale. Specifically, for the toxic token insertion, we generate a list of 200 toxic tokens and insert them at random positions in the watermarked output. For the fluent inaccurate editing, we edit the watermarked sentence by querying GPT4 using the prompt *"Modify less than 3 words in the following sentence and make it inaccurate or have opposite meanings."* Unless otherwise specified, in the evaluations of this work, we utilize 500 prompts data from OpenGen [18] dataset, and query the watermarked language models (LLAMA-2-7B [37] and OPT-1.3B [44]) to generate the watermarked outputs. We evaluate three SOTA watermarks including KGW [16], Unigram [45], and Exp [19], using the default watermarking hyperparameters. In our experiments, we default to a maximum of 200 new tokens for KGW and Unigram, and 70 for Exp, due to its complexity in the watermark detection. 70 is also the maximum number of tokens the authors of Exp evaluated in their paper [19].

**Evaluation Result.** We report the maximum portion of the inserted toxic tokens relative to the original watermarked sentence length on LLAMA-2-7B model in Fig. 1a. We also present the confidence of the OpenAI moderation model [29] in identifying the content as violating their usage policy [30] due to the inserted toxic tokens in Fig. 1a. Our findings show that we can insert a significant number of toxic tokens into content generated by all the robust watermarking schemes, with a median portion higher than 20%, i.e., for a 200-token sentence, the attacker can insert a median of 40 tokens into it. These toxic sentences are then identified as violating OpenAI policy rules with high confidence scores, whose median is higher than 0.8 for all the watermarking schemes we study. The average confidence scores for content before attack are around 0.01. The empirical data on the maximum portion of inserted toxic tokens aligns with our analysis in Appendix D. We further validate this analysis in Fig. 5 of Appendix E, showing that attackers can insert nontrivial portions of toxic tokens into the watermarked text to launch piggyback spoofing attacks. Notably, the more robust the watermark is, the more tokens can effectively be inserted. We present the results on OPT-1.3B in Appendix G.

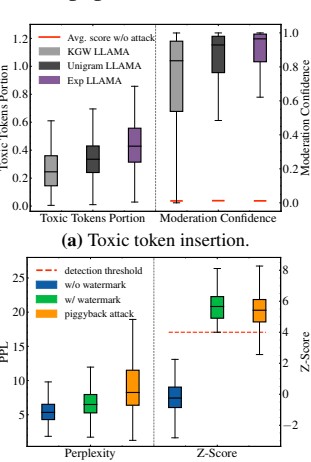

**(a)** Toxic token insertion.

**(b)** Fluent inaccurate editing.

**Figure 1:** Piggyback spoofing of robust watermarks. (a) We can insert a large number of toxic tokens in robustly watermarked text without changing the watermark detection result, resulting in text that is likely to be identified as toxic. (b) We can use GPT4 to automatically modify watermarked text, making it appear inaccurate while retaining fluency.

In Fig. 1b, we report the PPL and watermark detection scores of the piggyback results on KGW and LLAMA-2-7B by the fluent inaccurate editing strategy. We show that we can successfully generate fluent results, with a slightly higher PPL. 94.17% of the piggyback results have a z-score higher than the default threshold 4. We randomly sample 100 piggyback results and manually check that most of them (92%) are fluent and have inaccurate or opposite content from the original watermarked content. See examples in Appendix F. The results show that we can generate watermarked, fluent, but inaccurate content at scale with an ASR higher than 90%.

## 4.2 Discussion

> **Guideline #1**
>
> Robust watermarks are inherently vulnerable to spoofing attacks and are not suitable as proof of content authenticity alone. To mitigate spoofing while preserving robustness, it may be necessary to combine additional measures such as signature-based fragile watermarks.

Our results highlight that piggyback spoofing attacks are easy to execute in practice. LLM watermarks typically do not consider such attacks during design and deployment, and existing robust

watermarks are inherently vulnerable to such attacks. We highlight the contradiction between the watermark robustness and the piggyback spoofing feasibility. We consider this attack to be challenging to defend against, especially considering examples such as those in Table 1 and Appendix F, where by only editing a single token, the entire content becomes incorrect. It is hard, if not impossible, to detect whether a particular token is from the attacker by using robust watermark detection algorithms. Thus, practitioners should weigh the risks of removal vs. piggyback spoofing attacks for the model at hand. A feasible strategy to mitigate spoofing attacks is by requiring proof of digital signatures on the LLM generated content. However, while an attacker without access to the private key cannot spoof, it is worth nothing that this strategy is still vulnerable to watermark-removal attacks, as a single editing can invalidate the original signature.

# 5 Attacking Stealing-Resistant Watermarks

As discussed in Sec. 2, many works have explored the possibility of launching watermark stealing attacks to infer the secret pattern of the watermark, which can then enable spoofing and removal attacks [34, 14, 10]. A natural and effective defense against watermark stealing is using *multiple watermark keys* during embedding, which is a common practice in cryptography and also suggested by prior watermarks and work in watermark stealing [16, 19, 14]. Unfortunately, we demonstrate that using multiple keys can in turn introduce new watermark-removal attacks.

In particular, SOTA watermarking schemes [16, 9, 5, 19, 45, 17] aim to ensure the watermarked text retains its high quality and the private watermark patterns are not easily distinguished by maintaining an "unbiasedness" property:

$$\mathbb{E}_{sk \in \mathcal{S}}(M_{\text{wm}}(\mathbf{x}, sk)) \approx_\epsilon M_{\text{orig}}(\mathbf{x}), \tag{4}$$

i.e., the expected distribution of watermarked output over the watermark key space $sk \in \mathcal{S}$ is close to the output distribution without a watermark, differing by a distance of $\epsilon$. We note that Exp [19] is "distortion free" for a single text sample, and KGW [16] and Unigram [45] slightly shift the watermarked distributions. We note that stealing attacks won't work on rigorously unbiased watermarks.

The insight of our proposed watermark-removal attack is that given the "unbiasedness" nature of watermarks and considering multiple keys may be used during watermark embedding, malicious users can estimate the output distribution without any watermark by querying the watermarked LLM multiple times using the same prompt. As this attack estimates the original, unwatermarked distribution, the quality of the generated content is preserved.

**Attack Procedure.** An attacker queries a watermarked model with an input $\mathbf{x}$ multiple times, observing $n$ subsequent tokens $\mathbf{x}_{t+1}$. This is easy for text completion model APIs, and chat model APIs can also be easily attacked by constructing a prompt to ask the chat model to complete a partial sentence without any prefix. The attacker then creates a frequency histogram of these tokens and samples according to the frequency. This sampled token matches the result of sampling on an unwatermarked output distribution with a nontrivial probability. Consequently, the attacker can progressively eliminate watermarks while maintaining a high quality of the synthesized content. We present a formal analysis of the number of required queries in Appendix H.

## 5.1 Evaluation

**Experiment Setup.** Our watermarks, models and datasets settings are the same as Sec. 4.1. We study the trade-off between resistance against watermark stealing and watermark-removal attacks by evaluating a recent watermark stealing attack [14]. In this attack, we query the watermarked LLM to obtain 2.2 million tokens in total to estimate the watermark pattern and then launch spoofing attacks using the estimated watermark pattern. We follow their assumptions that the attacker can access the unwatermarked tokens' distribution. In our watermark removal attack, we consider that the attacker has observations with different keys. We evaluate the detection scores (z-score or p-value) and the output perplexity (PPL, evaluated using GPT3 [31]). The detection algorithm returns the maximum detection score across all the keys, which increases the expectation of unwatermarked detection results. Thus, we set the detection thresholds for different keys to keep the false positive rates (FPR) below 1e-3 and report the attack success rates (ASR). We use default watermark hyperparameters.

**Evaluation Result.** As shown in Fig. 2a, using multiple keys can effectively defend against watermark stealing attacks. With a single key, the ASR is 91%, which matches the results reported in [14]. We observe that using three keys can effectively reduce the ASR to 13%, and using more

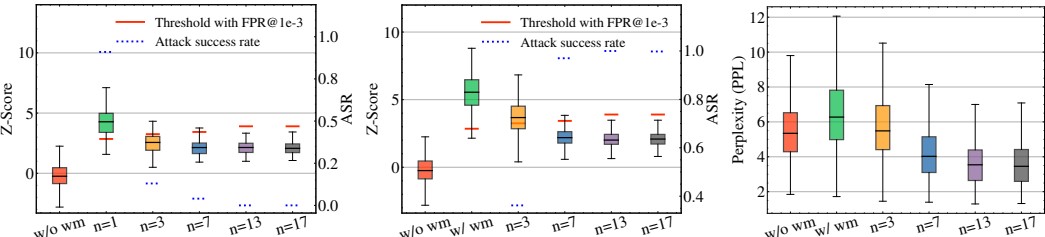

**(a)** Z-Score and attack success rate (ASR) of watermark stealing [14].

**(b)** Z-Score and attack success rate (ASR) of watermark-removal.

**(c)** Perplexity (PPL) of watermark-removal.

**Figure 2:** Spoofing attack based on watermark stealing [14] and watermark-removal attacks on KGW watermark and LLAMA-2-7B model with different number of watermark keys $n$. Higher z-score reflects more confidence in watermarking and lower perplexity indicates better sentence quality. The attack success rates are based on the threshold with FPR@1e-3.

than 7 keys, the ASR of the watermark stealing is close to zero. However, using more keys also makes the system vulnerable to our watermark-removal attacks as shown in Fig. 2b. When we use more than 7 keys, the detection scores of the content produced by our watermark removal attacks closely resemble those of unwatermarked content and are much lower than the detection thresholds, with ASRs higher than 97%. Fig. 2c suggests that using more keys improves the quality of the output content. This is because, with a greater number of keys, there is a higher probability for an attacker to accurately estimate the unwatermarked distribution, which is consistent with our analysis in Appendix H. We observe that in practice, 7 keys suffice to produce high-quality content comparable to the unwatermarked content. These observations remain consistent across various watermarking schemes and models; for additional results see Appendix J. We note that the numbers are not exactly the same as [14], as we consider a more realistic attacker with less queries to the watermarked LLM.

## 5.2 Discussion

> **Guideline #2**
>
> Using a larger number of watermarking keys can defend against watermark stealing attacks, but increases vulnerability to watermark-removal attacks. Limiting users' query rates can help to mitigate both attacks.

Many prior works have suggested using multiple keys to defend against watermark stealing attacks. However, in this study, we reveal that a conflict exists between improving resistance to watermark stealing and the feasibility of removing watermarks. Our evaluation results show that finding a "sweet spot" in terms of the number of keys to use to mitigate both the watermark stealing and the watermark-removal attacks is not trivial. For example, our watermark-removal attack achieves a high ASR of 36.2% just using three keys, and the corresponding watermark stealing-based spoofing's ASR is 13.0%. Using more keys can decrease the watermark stealing-based spoofing's ASR, but at the cost of making the system more vulnerable to watermark removal and vice-versa. We note that the ASRs with three keys are not negligible, thus limiting the ability of potentially malicious users is necessary in practice to mitigate these attacks. As a practical defense, we evaluate watermark stealing with various query limits on the watermarked LLM, and found that the ASR can be significantly reduced by limiting the attacker's query rate. Detailed results can be found in Appendix J. Given the trade-off that exists, we suggest that LLM service providers consider "defense-in-depth" techniques such as anomaly detection, query rate limiting, and user identification verification.

## 6 Attacking Watermark Detection APIs

It is still an open question whether watermark detection APIs should be made publicly available to users. Although this makes it easier to detect watermarked text, it is a commonly acknowledged that it will make the system vulnerable to attacks [1] given the existence of oracle attacks [3, 6, 22, 15]. Here, we study this statement more precisely by examining the specific risk trade-offs that exist, as well as introducing a novel defense that may make the public detection API more feasible in practice. In the following sections, we first introduce attacks that exploit the APIs and then propose suggestions and defenses to mitigate these attacks.

### 6.1 Attack Procedures

**Watermark-Removal Attack.** For the watermark-removal attack, we consider an attacker who has access to the target watermarked LLM's API, and can query the watermark detection results. The at-

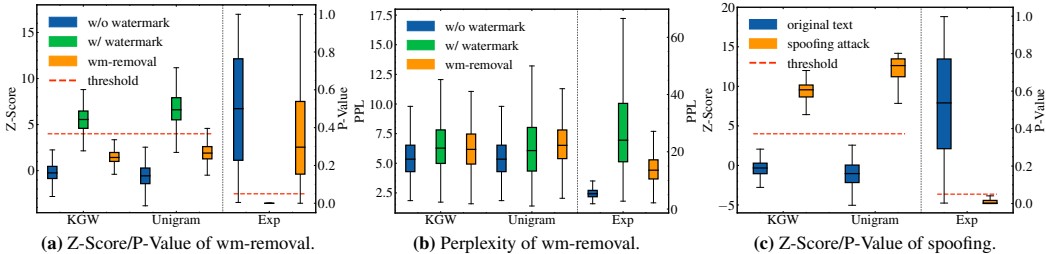

**(a)** Z-Score/P-Value of wm-removal.   **(b)** Perplexity of wm-removal.   **(c)** Z-Score/P-Value of spoofing.

**Figure 3:** Attacks exploiting detection APIs on LLAMA-2-7B model.

tacker feeds a prompt into the watermarked LLM, which generates the response in an auto-regressive manner, similar to how LLMs generate sentences. That is, the attacker will select each token based on the prior tokens and the detection results. For the token $\mathbf{x}_i$ the attacker will generate a list of possible replacements for $\mathbf{x}_i$. This list can be generated by querying the watermarked LLM, querying a local model, or simply returned by the watermarked LLM. In this work, we choose the third approach because of its simplicity and guarantee of synthesized sentences' quality. This is a common assumption made by prior works [25], and such an API is also provided by OpenAI ($\text{top\_logprobs} = 5$), which can benefit the normal users in understanding the model confidence, debugging and analyzing the model's behavior, customizing sampling strategies, etc. Consider that the top $L = 5$ tokens and their probabilities are returned to the attackers. The probability that the attacker can find an unwatermarked token in the token candidates' list of length $L$ is $1 - \gamma^L$ for KGW and Unigram, which becomes sufficiently large given $L = 5$ and $\gamma = 0.5$. The attacker will query the detection using these replacements and sample a token based on their probabilities and detection scores to remove the watermark while preserving a high output quality.

**Spoofing Attack.** Spoofing attacks follow a similar procedure where the attacker can generate (harmful) content using a local model. When sampling the tokens, instead of selecting those that yield low confidence scores as in removal attacks, the attacker will choose tokens that have higher confidence scores upon watermark detection queries. Thanks to the robustness of the LLM watermarks, attackers don't need to ensure every single token carries a watermark; only that the overall detection confidence score surpasses the threshold, thereby treating synthesized content as if generated by the watermarked LLM.

### 6.2   Evaluation

**Experiment Setup.**   We use the same evaluation setup as in Sec. 4.1 and Sec. 5.1. We evaluate the detection scores for both the watermark-removal and the spoofing attacks. We also report the number of queries to the detection API. Furthermore, for the watermark-removal attack, where the attackers care more about the output quality, we report the output PPL. For spoofing attacks, the attackers' local models are LLAMA-2-7B and OPT-1.3B.

**Evaluation Result.**   As shown in Fig. 3a and Fig. 3b, watermark-removal attacks exploiting the detection API significantly reduce detection confidence while maintaining high output quality. For instance, for the KGW watermark on LLAMA-2-7B model, we achieve a median z-score of $1.43$, which is much lower than the threshold $4$. The PPL is also close to the watermarked outputs ($6.17$ vs. $6.28$). We observe that the Exp watermark has higher PPL than the other two watermarks. This is because that Exp watermark is deterministic, while other watermarks enable random sampling during inference. Our attack also employs sampling based on the token probabilities and detection scores, thus we can improve the output quality for the Exp watermark.

|  | wm-removal | | spoofing | |
|---|---|---|---|---|
|  | ASR | #queries | ASR | #queries |
| KGW | 1.00 | 2.42 | 0.98 | 2.95 |
| Unigram | 0.96 | 2.66 | 0.98 | 2.96 |
| Exp | 0.96 | 1.55 | 0.85 | 2.89 |

**Table 2:** The attack success rate (ASR), and the average query numbers per token for the watermark-removal and spoofing attacks exploiting the detection API on LLAMA-2-7B model.

The spoofing attacks also significantly boost the detection confidence even though the content is not from the watermarked LLM, as depicted in Fig. 3c. We report the attack success rate (ASR) and the number of queries for both of the attacks in Table 2. The ASR quantifies how much of the generated content surpasses or falls short of the detection threshold. These attacks use a reasonable number of queries to the detection API and achieve high success rate, demonstrating practical feasibility. We observe consistent results on OPT-1.3B, please see Appendix K.

## 6.3 Defending Detection with Differential Privacy

In light of the issues above, we propose an effective defense using ideas from differential privacy (DP) [7] to counteract detection API based spoofing attacks. DP adds random noise to function results evaluated on private dataset such that the results from neighbouring datasets are indistinguishable. Similarly, we consider adding Gaussian noise to the distance score in the watermark detection, making the detection $(\epsilon, \delta)$-DP [7], and ensuring that attackers cannot tell the difference between two queries by replacing a single token in the content, thus increasing the hardness of launching the attacks. Considering an attacker can average multiple query results to reduce noise and estimate original scores without DP protection, we propose to calculate the noise based on the random seed generated by a pseudorandom function (PRF) with the sentence to be detected as the input. Specifically, $\text{seed} = \text{PRF}_{sk}(\mathbf{x})$, where $sk$ is the secret key held by the detection service. The users without the secret key cannot reverse or reduce the noise in the detection score. Thus, we can successfully mitigate the noise reduction via averaging multiple query results without comprising on utility or protection of the DP defense. In the following, we evaluate the utility of the DP defense and its performance in mitigating the spoofing attacks.

**Experiment Setup.** Firstly, we assess the utility of DP defense by evaluating the accuracy of the detection under various noise scales. Next, we evaluate the efficacy of the spoofing against DP detection defense using the same method as in Sec. 6.1. We select the optimal noise scale that provides best defense while keeping the drop in accuracy within $2\%$. We note that for KGW and Unigram watermarks, we add noise to the z-scores. Sensitivity varies with sentence length (e.g., $\Delta = \frac{h+1}{\sqrt{\gamma(1-\gamma)l}}$ for replacement editing, where $l$ is the sentence length, $h$ is the context width of the watermark, and $\gamma$ is the portion of the tokens in green list). The actual noise scale is proportional to $\sigma\Delta$.

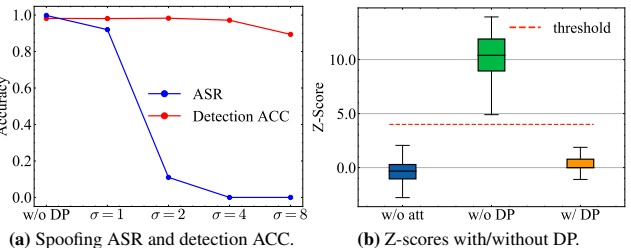

**(a)** Spoofing ASR and detection ACC. **(b)** Z-scores with/without DP.

**Figure 4:** Evaluation of DP detection on KGW watermark and LLAMA-2-7B model. **(a).** Spoofing attack success rate (ASR) and detection accuracy (ACC) without and with DP watermark detection under different noise parameters. **(b).** Z-scores of original text without attack, spoofing attack without DP, and spoofing attacks with DP. We use the best $\sigma = 4$ from **(a)**.

**Evaluation Result.** As shown in Fig. 4a, with a noise scale of $\sigma = 4$, the DP detection's accuracy drops from the original $98.2\%$ to $97.2\%$ on KGW and LLAMA-2-7B, while the spoofing ASR becomes $0\%$ using the same attack procedure as Sec. 6.1. The results are consistent for Unigram and Exp watermarks and OPT-1.3B model as shown in Appendix L, which illustrates that the DP defense has a great utility-defense trade-off, with a negligible accuracy drop and significantly mitigates the spoofing attacks.

## 6.4 Discussion

> **Guideline #3**
>
> Public detection APIs can enable both spoofing and removal attacks. To defend against these attacks, we propose a DP-inspired defense, which combined with techniques such as anomaly detection, query rate limiting, and user identification verification can help to make public detection more feasible in practice.

The detection API, available to the public, aids users in differentiating between AI and human-created materials. However, it can be exploited by attackers to gradually remove watermarks or launch spoofing attacks. We propose a defense utilizing the ideas in differential privacy, which significantly increases the difficulty for spoofing attacks. However, this method is less effective against watermark-removal attacks that exploit the detection API because attackers' actions will be close to random sampling, which, even though with less success rates, remains an effective way of removing watermarks. Therefore, we leave developing a more powerful defense mechanism against watermark-removal attacks exploiting detection API as future work. Additionally, we note that the attacker may increase the sensitivity of the input sentences by substituting multiple tokens and infer whether these tokens are in the green list or not to launch the spoofing attack, but this will require much more queries to the detection API. We recommend companies providing detection services

adopt a defense-in-depth approach [2, 8]. For instance, they should detect and curb malicious behavior by limiting query rates from potential attackers, and also verify the identity of the users to protect against Sybil attacks.

# 7   Discussion, Limitation & Future Work

**Generalizability of our attacks.** We focus on three SOTA PRF-based robust watermarks, which are a natural set to explore given their popularity and formal guarantees. There are other watermarks like the semantics-based watermarks [23, 33]. While attacking semantics-based watermarks is outside the scope of our study, we deem this an interesting future direction to explore.

Recently, researchers have also proposed signature-based publicly detectable watermarks [9] to mitigate the spoofing attacks by exploiting robustness. Unlike the watermarks we study, these watermarks usually have weaker robustness guarantees, which further highlights the trade-offs between robustness and vulnerability to spoofing attacks, as we have discussed in Sec. 4.

Our findings, such as exploiting robustness properties and publicly available detection APIs, can also be generalized to image watermarks [39, 42]. The attackers must integrate domain-specific constraints to ensure that the generated sentences or images are meaningful and high-quality. We deem studying the fundamental trade-offs for image watermarks a promising future direction.

**Trade-offs of watermark context width.** There are two effective strategies to mitigate the watermark stealing attacks for the KGW watermark [16]: 1) using a larger context width $h$ and 2) using multiple watermark keys. In this work (Sec. 5), we primarily explore the fundamental trade-offs in using multiple watermark keys, which prior work has underexplored. Trade-offs in context widths were discussed in previous work [14, 16, 45]. Using a larger $h$ increases resistance against watermark stealing but reduces robustness. Recent work [14] shows successful watermark stealing even with $h = 4$. Using multiple keys, as shown in Sec. 5 of our paper, mitigates stealing attacks but introduces new attack vectors of watermark removal. Our attacks will work under different choices of context width, as we exploit properties or design choices orthogonal to the context width. To demonstrate this point, we provide more experimental results in Appendix M.

**The influence of how detection proceeds with multiple keys.** For the scenarios where multiple keys are used, we consider the detector using min/max aggregation to obtain the detection score. More robust aggregations exist including the Harmonic mean p-value [40]. We note that our watermark-removal attack exploiting the use of multiple keys is not dependent on the aggregation method as we do not rely on the server's watermark detection. However, the trade-off analysis and the sweet spot for the number of keys may slightly change given the different detection performance.

**Changing to p-values in KGW and Unigram.** P-values are used for Exp [19] watermark in our paper, and the observations are consistent with KGW [16] and Unigram [45]. We expect no impact on results from this change since p-values are monotonic to z-scores. To ease the figures' presentation, we adopt the z-statistics in the main paper, we present more results of using p-values in Appendix M.

# 8   Conclusion

In this work, we reveal new attack vectors that exploit common features and design choices of LLM watermarks. In particular, while these design choices may enhance robustness, resistance against watermark stealing attacks, and public detection ease, they also allow malicious actors to launch attacks that can easily remove the watermark or damage the model's reputation. Based on the theoretical and empirical analysis of our attacks, we suggest guidelines for designing and deploying LLM watermarks along with possible defenses to establish more reliable LLM watermark systems.

## Acknowledgements

This work was supported in part by the National Science Foundation grants IIS2145670, CCF2107024, CNS2326312 and funding from Amazon, Apple, Google, Intel, Meta, and the CyLab Security and Privacy Institute. Any opinions, findings and conclusions or recommendations expressed in this material are those of the author(s) and do not necessarily reflect the views of any of these funding agencies.

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

## A Broader Impacts

Our work studies the security implications of common LLM watermarking design choices. By developing realistic attacks and defenses and a simple set of guidelines for watermarking in practice, we aim for the work to serve as a resource for the development of secure LLM watermarking systems. Of course, by outlining such attacks, there is a risk that our work may in fact increase the prevalence of watermark removal or spoofing attacks performed in practice. We believe that this is nonetheless an important step towards educating the community about potential risks in watermarking systems and ultimately creating more effective defenses for secure LLM watermarking.

More generally, our work shows that a number of trade-offs exist in LLM watermarking (e.g., between utility, usability, robustness, and susceptibility to removal or spoofing attacks). The guidelines we propose provide rough proposals for considering these trade-offs, but we note that how to best navigate each trade-off will depend on the application at hand. Considering strategies to best navigate this space for specific LLM watermarking applications is an important direction of future study.

## B Attacker's Motivation

We present two practical scenarios to motivate *watermark-removal* attacks: *(i)* A student or a journalist uses high-quality watermarked LLMs to write articles, but wants to remove the watermark to claim originality. *(ii)* A malicious company offering LLM services for clients, instead of developing their own LLMs, simply queries a watermarked LLM from a victim company and removes the watermark, potentially infringing upon IP rights of the victim company.

In *piggyback and spoofing* attacks, an attacker can damage the reputation of a victim company offering an LLM service. For example: *(i)* The attacker can use a spoofing attack to generate fake news or incorrect facts and post them on social media. By claiming the material is generated by the LLM from the benign company, the attacker can damage the reputation of the company and their model. *(ii)* The attacker can use the spoofing attack to inject malicious code into some public software. The code has the benign company's watermark embedded, and the benign company may thus be at fault and have to bear responsibility for the actions.

## C Watermarking Schemes & Hyper-Parameters

In this section, we introduce the three watermarking schemes we evaluate in the paper—KGW [16], Unigram [45], and Exp [19]. We also introduce the perplexity, a metric to evaluate the sentence quality.

**KGW.** In the KGW watermarking scheme, when generating the current token $\mathbf{x}_{t+1}$, all the tokens in the vocabulary is pseudorandomly shuffled and split into two lists—the green list and the red list. The random seed used to determine the green and red lists is computed by a watermark secret key $sk$ and the prior $h$ tokens $\mathbf{x}_{t-h-1}||\cdots||\mathbf{x}_t$ using pseudorandom functions (PRFs):

$$\text{SEED} = F_{sk}(\mathbf{x}_{t-h-1}||\cdots||\mathbf{x}_t),$$

where $h$ is the context width of the watermark. We note that the choice of $h$ has minor influence on our attacks or defenses, as our algorithms are not dependent on $h$. Here we use their original algorithm with $h = 1$. Then, the seed is used to split the vocabulary into the green and red lists of tokens, with $\gamma$ portion of tokens in the green list:

$$L_{\text{green}}, L_{\text{red}} = \text{Shuffle}(\mathcal{V}, \text{SEED}, \gamma)$$

Then, KGW generates a binary watermark mask vector for the current token prediction, which has the same size as the vocabulary. All the tokens in the green list $L_{\text{green}}$ have value 1 in the mask, and all the tokens in the red list have value 0 in the mask:

$$\text{MASK} = \text{GenerateMask}(L_{\text{green}}, L_{\text{red}})$$

To embed the watermark, KGW add a constant to the logits of the LLM's prediction for token $\mathbf{x}_{t+1}$:

$$\text{WATERMARKEDPROB} = \text{Softmax}(\text{logits} + \delta \times \text{MASK}),$$

where the logits is from the LLM, and the $\delta$ is the watermark strength. Then the LLM will sample the token $\mathbf{x}_{t+1}$ according to the watermarked probability distribution.

The detection involves computing the z-score:

$$z = \frac{g - \gamma l}{\sqrt{\gamma(1 - \gamma)l}},$$

where $g$ is the number of tokens in the green list, $l$ is the total number of tokens in the input token sequence, and $\gamma$ is the portion of the vocabulary tokens in the green list. Similar to the watermark embedding, the green and red lists for each token position are determined by watermark secret key and the token prior to the current token in the input token sequence.

**Unigram.** Similar to KGW, Unigram also splits the vocabulary into green and red lists and prioritize the tokens in the green list by adding a constant to the logits before computing the softmax. The difference is that Unigram uses global red and green lists instead of computing the green and red lists for each token. That is, the seed to shuffle the list is only determined by the watermark secret key and generated by a Pseudo-Random Generator (PRG):

$$\text{SEED} = G(sk)$$

Then, similar to KGW, the seed is used to split the vocabulary into the green and red lists of tokens, with $\gamma$ portion of tokens in the green list:

$$L_{\text{green}}, L_{\text{red}} = \text{Shuffle}(\mathcal{V}, \text{SEED}, \gamma)$$

The watermark embedding and detection procedures are the same as KGW: Unigram first compute the watermark mask:

$$\text{MASK} = \text{GenerateMask}(L_{\text{green}}, L_{\text{red}})$$

And then embed the watermark by perturbing the logits of the LLM outputs:

$$\text{WATERMARKEDPROB} = \text{Softmax}(\text{logits} + \delta \times \text{MASK}),$$

where the logits is from the LLM, and the $\delta$ is the watermark strength. Then the LLM will sample the token $\mathbf{x}_{t+1}$ according to the watermarked probability distribution.

The detection also computes the z-score:

$$z = \frac{g - \gamma l}{\sqrt{\gamma(1 - \gamma)l}},$$

where $g$ is the number of tokens in the green list, $l$ is the total number of tokens in the input token sequence, and $\gamma$ is the portion of the vocabulary tokens in the green list. According to the analysis in [45] and also consistent with our results in Sec. 4.1, by decoupling the green and red lists splitting with the prior tokens, Unigram is twice as robust as KGW. But it's more likely to leak the pattern of the watermarked tokens given that it uses a global green-red list splitting.

**Exp.** The Exp watermarking scheme from [19] is an extension of [1]. Instead of using a single key as in KGW and Unigram, the usage of multiple watermark keys is inherent in Exp to provide the distortion-free guarantee. Each key is a vector of size $|\mathcal{V}|$ with values uniformly distributed in $[0, 1]$. That is, $sk = \xi_1, \xi_2, \cdots, \xi_n$, where $\xi_k \in [0, 1]^{|\mathcal{V}|}, k \in [n]$, and $n$ is the length of the watermark keys, default to $256$.

For the prediction of the token $\mathbf{x}_{t+1}$, Exp firstly collects the output probability vector $\mathbf{p} \in [0, 1]^{|\mathcal{V}|}$ from the LLM. A random shift $r \xleftarrow{\$} [n]$ is sampled at the beginning of receiving the prompt. Then the token $\mathbf{x}_{t+1}$ is sampled using the Gumbel trick [11]:

$$\mathbf{x}_{t+1} = \arg\max_i \ (\xi_{k,i})^{1/\mathbf{p}_i},$$

where $k = r + t + 1 \bmod n$, i.e., each position uses a different watermark key which determines the uniform distribution sampling used in the Gumbel trick sampling. This method guarantees that the output distribution is distortion-free, whose expectation is identical to the distribution without watermark given sufficiently large $n$.

The watermark detection also computes test statistics. The basic test statistics is:

$$\phi = \sum_{t=1}^{l} -\log(1 - \xi_{k,\mathbf{x}_t}),$$

where $k = t \bmod n$. And Exp computes the minimum Levenshtein distance using the basic test statistic as a cost (see Sec. 2.4 in [19]).

Instead of using single keys as KGW and Unigram, Exp uses multiple keys and incorporates Gumbel trick to rigorously provide distortion-free (unbiased) guarantee, whose expected output distribution over the key space is identical to the unwatermarked distribution.

**Sentence Quality.** Perplexity (PPL) is one of the most common metrics for evaluating language models. It can also be utilized to measure the quality of the sentences [45, 16] based on the oracle of high-quality language models. Formally, PPL returns the following quality score for an input sentence **x**:

$$\text{PPL}(\mathbf{x}) = \exp\{-\frac{1}{t}\sum_{i=1}^{t}\log[\Pr(\mathbf{x}_i|\mathbf{x}_0,\cdots\mathbf{x}_{i-1})]\} \tag{5}$$

In our evaluation, we utilize the GPT3 [31] as the oracle model to evaluate sentence quality.

**Setups and Hyper-Parameters.** For KGW [16] and Unigram [45] watermarks, we utilize the default parameters in [45], where the watermark strength is $\delta = 2$, and the green list portion is $\gamma = 0.5$. We employ a threshold of $T = 4$ for these two watermarks with a single watermark key. For the scenarios where multiple keys are used, we calculate the thresholds to guarantee that the false positive rates (FPRs) are below 1e-3. For the Exp watermark (refered to as Exp-edit in [19]), we use the default parameters, where the watermark key length is $n = 256$ and the block size $k$ is default to be identical to the token length. We set the p-value threshold for Exp to $0.05$ in our experiments.

For the spoofing attack exploiting detection APIs, we obtain the first three tokens with the highest probabilities from the unwatermarked LLM, and for the removal attack exploiting detection APIs, we obtain the first five tokens with the highest probabilities from the watermarked LLM. For watermark removal attacks exploiting detection APIs on KGW and Unigram, we increase the probabilities of the tokens that have the smallest detection confidence, and then sample from the modified probability distribution. For watermark removal attacks exploiting detection APIs on Exp, we simply sample the token that has the maximum p-value, but we will skip the tokens that have low probabilities (lower than $0.15$) when the detection p-value is high (higher than $0.1$). The different setup for the Exp watermark is required to ensure that we can produce high-quality sentences. For watermark spoofing attacks that exploit detection APIs, we sample the token that has the highest detection confidence for KGW, Unigram, and Exp watermarks.

We conduct the experiments on a cluster with 8 NVIDIA A100 GPUs, AMD EPYC 7763 64-Core CPU, and 1TB memory.

## D    Attack Feasibility Analysis of Piggyback Spoofing Exploiting Robustness

We study the bound on the maximum number of tokens that are allowed to be inserted or edited in a watermarked sentence, and we present the following theorem on Unigram watermark [45] due to its clean robustness guarantee:

**Theorem 1** (Maximum insertion portion). *Consider a watermarked token sequence $\mathbf{x}$ of length $l$. The Unigram watermark z-score threshold is $T$, the portion of the tokens in the green list is $\gamma$, the detection z-score of $\mathbf{x}$ is $z$, and the number of inserted tokens is $s$. Then, to guarantee the expected z-score of the edited text is greater than $T$, it suffices to guarantee $\frac{s}{l} \leq \frac{z^2 - T^2}{T^2}$.*

*Proof.* Recall that the watermarking schemes' detections usually involve computing the statistical testing. Unigram splits the vocabulary into two lists—the green list and the red list. It prioritizes the tokens in the green list during watermark embedding, and the detection computes the z-score:

$$z = \frac{g - \gamma l}{\sqrt{\gamma(1-\gamma)l}},$$

where $g$ is the number of tokens in the green list, $l$ is the total number of tokens in the input token sequence, and $\gamma$ is the portion of the vocabulary tokens in the green list. Let the number of the inserted toxic tokens be $s$. Since toxic tokens are independent of the secret key $sk$, the expected new z-score $z'$ is:

$$\mathbb{E}(z') = \frac{g + \gamma s - \gamma(l+s)}{\sqrt{\gamma(1-\gamma)(l+s)}} = z\sqrt{\frac{l}{l+s}},$$

To guarantee that $\mathbb{E}(z') \geq T$, it suffices to guarantee

$$\frac{s}{l} \leq \frac{z^2 - T^2}{T^2}$$

□

Different from the analysis in the Unigram paper on how the z-score changes given a specific number of edits, we have a tight bound on the maximum possible number of edits, which is also more straightforward for the attack feasibility analysis. According to Theorem 1, as long as the number of toxic tokens inserted is bounded by $l\frac{z^2-T^2}{T^2}$, the attacker can execute a piggyback attack to generate toxic content with the target watermark embedded. The editing distance bound (Def. 3) for a sentence is $\epsilon = l\frac{z^2-T^2}{T^2}$. A stronger watermark makes piggyback spoofing attacks easier by allowing more toxic tokens to be inserted. This conclusion applies universally to all robust watermarking schemes. This is a fundamental design trade-off: if a watermark is robust, such spoofing attacks are inevitable and may be extremely difficult to detect, as even one toxic token can render the entire content harmful or inaccurate.

# E    Validation of Theorem 1

In this section, we validate Theorem 1 by using watermarked texts of varying lengths $l$ and z-scores $z$ to study the relationship between $\frac{s}{l}$ and $\frac{z^t-T^2}{T^2}$ of Unigram watermark. The results are shown in Fig. 5. As anticipated, 85.78% of the maximum allowable tokens to be inserted into the watermarked content satisfy Theorem 1. Given that this equation analyzes expected $s/l$, a small portion of outliers is reasonable. We primarily visualize this result for Unigram due to its clean robustness guarantee. Other watermarks can also reach similar conclusions, but their bounds on $s$ are either complex [16] or lack a closed form [19], making them difficult to visualize. Our empirical findings in Fig. 1 sufficiently prove an attacker can insert nontrivial portions of toxic or incorrect tokens into the watermarked text to launch the spoofing attack, which can be generalized across all robust watermarking schemes.

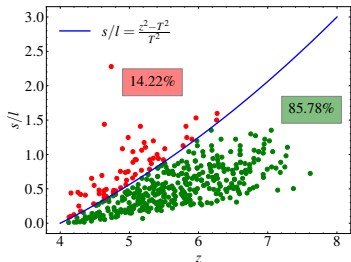

**Figure 5:** The relationship between $s/l$ and $z$. The data points are evaluated on Unigram using LLAMA-2-7B and 500 samples from OpenGen dataset.

# F    Piggyback Attack Examples

Here we present more piggyback attack results using the edition strategy by querying GPT4 using the prompt *"Modify less than 3 words in the following sentence and make it inaccurate or have opposite meanings."* The attack is launched on KGW watermark and LLAMA-2-7B model.

---

**Watermarked content, z-score: 4.93, PPL: 4.61**

*Earth has a history of* 4.5 billion years and humans have been around for 200,000 years. Yet humans have been using computers for just over **70** years and even then the term was first used in **1945**. In the age of technology, we are still just getting started. The first computer, ENIAC (Electronic Numerical Integrator And Calculator), was built at the University of Pennsylvania between 1943 and 1946. The ENIAC took up 1800 sq ft and had 18,000 vacuum tube and mechanical parts. The ENIAC was used for mathematical calculations, ballistics, and code breaking. The ENIAC was 1000 times **faster** than any other calculator of the time. The first computer to run a program was the Z3, built by Konrad Zuse at his house.

---

**Piggyback attack, z-score: 4.36, PPL: 5.68**

*Earth has a history of* 4.5 billion years and humans have been around for 200,000 years. Yet humans have been using computers for just over **700** years and even then the term was first used in **1445**. In the age of technology, we are still just getting started. The first computer, ENIAC (Electronic Numerical Integrator And Calculator), was built at the University of Pennsylvania between 1943 and 1946. The ENIAC took up 1800 sq ft and had 18,000 vacuum tube and mechanical parts. The ENIAC was used for mathematical calculations, ballistics, and code breaking. The ENIAC was 1000 times **slower** than any other calculator of the time. The first computer to run a program was the Z3, built by Konrad Zuse at his house.

**Watermarked content, z-score: 4.57, PPL: 3.51**

*Harry Potter is* a 2001 film adaptation of J.K. Rowling's best-selling novels. In the year **1991**, Harry Potter is rescued from his aunt and uncle's home and taken to Hogwarts, a school for magic. During the course of his first year, Harry makes friends and learns about the evil Lord Voldemort and his attempts to kill him. The film was directed by Chris Columbus, and stars Daniel Radcliffe as Harry Potter, with Richard Harris, Maggie Smith, Robbie Coltrane, Alan Rickman, Emma Watson, Rupert Grint, Gary Oldman, and Julie Walters in supporting roles. In the first film, Harry Potter is taken to Hogwarts School of Witchcraft and Wizardry, where he makes friends and discovers that Lord Voldemort wants to kill him.

**Piggyback attack, z-score: 4.47, PPL: 3.78**

*Harry Potter is* a 2001 film adaptation of J.K. Rowling's best-selling novels. In the year **2991**, Harry Potter is rescued from his aunt and uncle's home and taken to Hogwarts, a school for magic. During the course of his first year, Harry makes friends and learns about the evil Lord Voldemort and his attempts to kill him. The film was directed by Chris Columbus, and stars Daniel Radcliffe as Harry Potter, with Richard Harris, Maggie Smith, Robbie Coltrane, Alan Rickman, Emma Watson, Rupert Grint, Gary Oldman, and Julie Walters in supporting roles. In the first film, Harry Potter is taken to Hogwarts School of Witchcraft and Wizardry, where he makes friends and discovers that Lord Voldemort wants to kill him.

**Watermarked content, z-score: 6.15, PPL: 4.29**

*Usain Bolt is* a 6'6", 200-pound sprinter from the island of Jamaica. He's also arguably the greatest athlete to ever walk the earth. In an era where many of us are glued to our phones, he has over 18 million followers on Instagram. The 31-year-old athlete is known for his incredible speed and endurance. He's won **eight** Olympic gold medals, including three in 2012 alone. He's also set world records in both the 100-meter dash and 200-meter dash. Bolt is the only person to ever hold the world record in both events at the same time, which he did twice. He's also the only person to ever hold the world record in both events at the same Olympics. Bolt is known for his incredible speed and endurance.

**Piggyback attack, z-score: 5.58, PPL: 4.46**

*Usain Bolt is* a 6'6", 200-pound sprinter from the island of Jamaica. He's also arguably the greatest athlete to ever walk the earth. In an era where many of us are glued to our phones, he has over 18 million followers on Instagram. The 31-year-old athlete is known for his incredible speed and endurance. He's won **three** Olympic gold medals, including three in 2012 alone. He's also set world records in both the 100-meter dash and 200-meter dash. Bolt is the only person to ever hold the world record in both events at the same time, which he did twice. He's also the only person to ever hold the world record in both events at the same Olympics. Bolt is known for his incredible speed and endurance.

**Watermarked content, z-score: 6.01, PPL: 6.68**

*The history of the modern airplane is* 100 years old this month. And yet it's not been 100 years since the Wright Brothers' first flight. The first airplane flight took place on Dec. 17, 1903. After three years of development, Orville and Wilbur Wright's first flight lasted only 12 seconds. But within a decade, the first airliner flew. In 1924, the Ford Motor Co. **flew** the first commercial plane on the U.S. East Coast. In the next year, the company built a 10-passenger airliner with passenger windows and seats and an aisle. The 10-seat plane was called the Model T, and Ford executives said it would have been better if the company made a 10-passenger car instead of a plane.

> **Piggyback attack, z-score: 5.03, PPL: 7.19**
>
> *The history of the modern airplane is* 100 years old this month. And yet it's not been 100 years since the Wright Brothers' first flight. The first airplane flight took place on Dec. 17, 1903. After three years of development, Orville and Wilbur Wright's first flight lasted only 12 seconds. But within a decade, the first airliner flew. In 1924, the Ford Motor Co. **never flew** the first commercial plane on the U.S. East Coast. In the next year, the company built a 10-passenger airliner with passenger windows and seats and an aisle. The 10-seat plane was called the Model T, and Ford executives said it would have been better if the company made a 10-passenger car instead of a plane.

# G   Additional Results of Piggyback Spoofing Attack

In Sec. 4, we present the piggyback spoofing attack using toxic token insertion strategy on LLAMA-2-7B model. Here, we present the results on OPT-1.3B model, which are consistent with LLAMA-2-7B model's results.

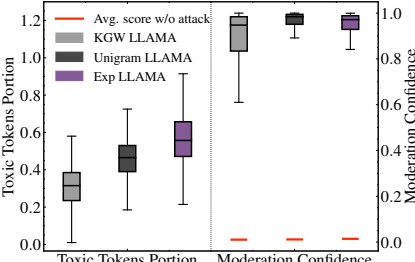

**Figure 6:** Piggyback spoofing of robust watermarks with toxic token insertion strategy on OPT-1.3B.

In Sec. 4, we present the fluent inaccurate editing strategy by querying the GPT4 on KGW watermark and LLAMA-2-7B model. Here we present more results of this strategy on all the three watermarks (KGW, Unigram, and Exp) and two models (LLAMA-2-7B and OPT-1.3B). The results are shown in Fig. 7, Fig. 8, and Fig. 9, which are consistent with our findings in Fig. 1, indicating that our piggyback spoofing attack can be generalized across various robust watermarks and models.

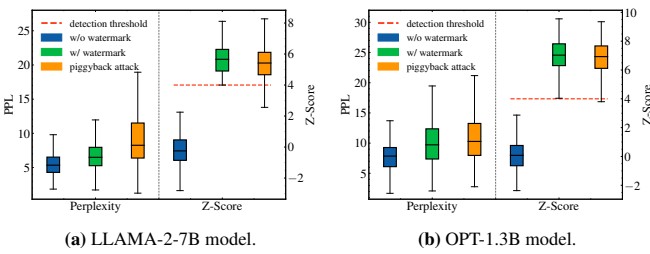

**(a)** LLAMA-2-7B model.      **(b)** OPT-1.3B model.

**Figure 7:** Fluent inaccurate editing strategy on KGW watermark and LLAMA-2-7B and OPT-1.3B models.

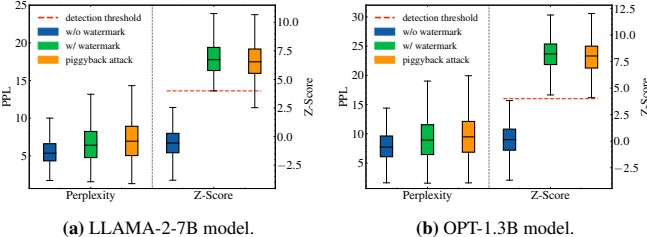

**(a)** LLAMA-2-7B model.      **(b)** OPT-1.3B model.

**Figure 8:** Fluent inaccurate editing strategy on Unigram watermark and LLAMA-2-7B and OPT-1.3B models.

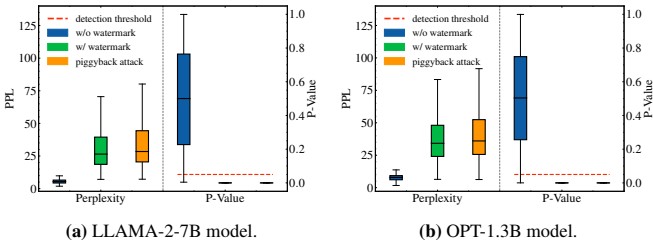

| (a) LLAMA-2-7B model. | (b) OPT-1.3B model. |

**Figure 9:** Fluent inaccurate editing strategy on Exp watermark and LLAMA-2-7B and OPT-1.3B models.

# H    Watermark Key Number Analysis for Watermark-Removal Attacks Exploiting the Use of Multiple Watermark Keys

Now we analyze the number of required queries under different keys to estimate the token with the highest probability without a watermark. We have the following probability bound for KGW and Unigram with the corresponding proof, and present the bound for Exp in Appendix I.

**Theorem 2** (Probability bound of unwatermarked token estimation). *Suppose there are $n$ observations under different keys, the portion of the green list in KGW or Unigram is $\gamma$. Then the probability that the most frequent token is the same as the original unwatermarked token is*

$$1 - \sum_{k=0}^{\lfloor n/2 \rfloor} \binom{n}{k} \gamma^k (1-\gamma)^{n-k} \times p(k), \tag{6}$$

*where $p(k) = 1 - \left( \sum_{m=0}^{k-1} \binom{n-k}{m} \gamma^m (1-\gamma)^{n-k-m} \right)^c$, $c$ is the number of other tokens whose watermarked probability can exceed that of the highest unwatermarked token.*

In a practical scenario where $n = 13, \gamma = 0.5$, and $c = 3$, Theorem 2 suggests that the attacker has a probability of $0.71$ in finding the token with the highest unwatermarked probability. This implies that we can successfully remove watermarks from over $71\%$ of tokens using a small number of observations under different keys ($n = 13$), yielding high-quality unwatermarked content.

*Proof.* Recall that KGW and Unigram randomly split the tokens in the vocabulary into the green list and the red list. We consider the greedy sampling, where the token with the highest (watermarked) probability is sampled. We have $n$ independent observations under different watermark keys. For each key, the token $\mathbf{x}_i$ with the highest unwatermarked probability is in the green list is $\gamma$. As long as $\mathbf{x}_i$ is the green list, the greedy sampling will always yield $\mathbf{x}_i$ since the watermarks add the same constant to all the tokens' loogits in the green list.

Thus, the probability that the most frequent token among these $n$ observations is $\mathbf{x}_i$ is at least:

$$1 - \sum_{k=0}^{\lfloor n/2 \rfloor} \binom{n}{k} \gamma^k (1-\gamma)^{n-k},$$

which is the probability that $\mathbf{x}_i$ is in the green list for at least half of the $n$ keys.

For another token $\mathbf{x}_j$ whose probability can exceed $\mathbf{x}_i$, if $\mathbf{x}_j$ is in the green list and $\mathbf{x}_i$ is in the red list. Then if $\mathbf{x}_i$ is in the green list for $k$ keys, the probability that $\mathbf{x}_j$ is in the green list for at least $k$ keys among the other $n - k$ keys is:

$$1 - \sum_{m=0}^{k-1} \binom{n-k}{m} \gamma^m (1-\gamma)^{n-k-m}$$

Consider we have $c$ such tokens having potential to exceed $\mathbf{x}_i$. Then at least one of the $c$ tokens is in the green list for at least $k$ keys among the other $n - k$ keys is:

$$1 - \left( \sum_{m=0}^{k-1} \binom{n-k}{m} \gamma^m (1-\gamma)^{n-k-m} \right)^c$$

Thus, with all the above analysis, we have that if there are $c$ tokens that have the potential to exceed the probability of the token with highest unwatermarked probability (i.e., $\mathbf{x}_i$), the probability that the most frequent token among the $n$ observations is the same as $\mathbf{x}_i$ is:

$$1 - \sum_{k=0}^{\lfloor n/2 \rfloor} \binom{n}{k} \gamma^k (1-\gamma)^{n-k} \times \left( 1 - \Big( \sum_{m=0}^{k-1} \binom{n-k}{m} \gamma^m (1-\gamma)^{n-k-m} \Big)^c \right),$$

which concludes the proof. $\qquad\square$

Here we consider that the watermarked LLM is utilizing greedy sampling. In practice, the greedy sampling might not be an optimal sampling strategy, but we note that it is extremely challenging to incorporate the multinomial sampling when analyzing the KGW and Unigram watermarks. Because KGW and Unigram add bias to the output logits, which will go through the softmax function to calculate the probabilities for the tokens. Given the softmax function is not unbiased, we cannot get a tight bound on its variance. Thus, we leave this part as a future direction to further incorporate multinomial sampling in the analysis. Nevertheless, our empirical results still show that the attackers can generate high-quality unwatermarked content when multinomial sampling is used. Also, our analysis on Exp watermark in Appendix I can naturally incorporate multinomial sampling.

## I   Probability Bound of Unwatermarked Token Estimation for Exp

In this section, we present and prove the probability bound of unwatermarked token estimation for the Exp watermark [19].

**Theorem 3** (Probability bound of unwatermarked token estimation for Exp). *Suppose there are $n$ observations under different keys, the highest probability for the unwatermarked tokens is $p$. Then the probability that the most frequently appeared token among the $n$ observations is the same as the original unwatermarked token with highest probability is:*

$$1 - \sum_{k=0}^{\lfloor n/2 \rfloor} \binom{n}{k} p^k (1-p)^{n-k} \tag{7}$$

*Proof.* The proof of Theorem 3 is straightforward. As we have introduced in Appendix C, the Exp watermark employs the Gumbel trick sampling [11] when embedding the watermark. Thus, the probability that we observe the token whose original unwatermarked probability is $p$ is exactly $p$ for each of the independent keys. Thus, if we make $n$ observations under different keys, then at least half of them yields the token with the highest original probability $p$ is:

$$1 - \sum_{k=0}^{\lfloor n/2 \rfloor} \binom{n}{k} p^k (1-p)^{n-k},$$

which concludes the proof. $\qquad\square$

## J   Additional Results of Watermark-Removal Attacks Exploiting the use of Multiple Watermark Keys

In this section, we provide more evaluation results of the watermark stealing [14] and our watermark-removal attacks exploiting the use of multiple watermark keys (see Sec. 5) on all the three watermarks (KGW, Unigram, and Exp) and two models (LLAMA-2-7B and OPT-1.3B). The results are shown in Fig. 11, Fig. 12, Fig. 13, Fig. 14, Fig. 15. For KGW watermark on OPT-1.3B model and Unigram watermark on LLAMA-2-7B and OPT-1.3B models, we have consistent observations with the KGW watermark on LLAMA-2-7B as we present in Sec. 5.1, demonstrating the effectiveness and generalizability of our attacks. For the Exp watermark, our results in Fig. 12 and Fig. 15 also show that the watermark can be easily removed using multiple queries to estimate the distribution of the unwatermarked tokens.

The results of the watermark stealing [14] on Unigram watermark and OPT-1.3B model are also consistent with our observations in Sec. 5. Using more keys can effectively mitigate the watermark stealing; however, it will make the system more vulnerable to our watermark removal attacks.

Throughout these experiments, we observe that using three keys is the optimal choice to defend against both attacks. However, the attack success rates with three keys are not negligible. Thus, consistent with our guidelines in Sec. 5, we highly recommend that the LLM service provider to simultaneously limit the ability of the potentially malicious users.

To further verify that the LLM service provider can mitigate the watermark stealing attacks by limiting the attacker's query rates, we present the stealing attack results with various numbers of queries on the KGW watermark and LLAMA-2-7B model using three keys in Fig. 10. The results show that by limiting the query rates of the attacker, the attack success rate of the watermark stealing attack can be significantly decreased. Thus, we recommend that the LLM service provider follow a "defense-in-depth" approach and utilize complementary techniques such as anomaly detection, query rate limiting, and user identification verification to mitigate stealing and removal attacks.

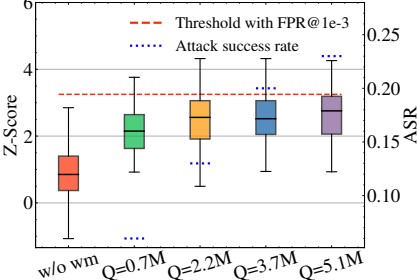

**Figure 10:** Watermark stealing attack [14] on KGW watermark and LLAMA-2-7B model using three keys with different numbers of attacker obtained tokens Q (in million). The attack success rates are based on the threshold with FPR@1e-3.

We note that the watermark stealing attacks do not work on the Exp watermark [19], as the use of a large number of watermark keys is inherent in their design, which defaults to 256. Thus, we omit the watermark stealing results on Exp, but we show that this watermark is inherently vulnerable to our watermark removal attack. From the results in Fig. 12 and Fig. 15, we conclude that using $n = 13$ queries, the resulting p-value is very close to that of the content without a watermark and is significantly different from the watermarked p-value, which shows that we can effectively remove the watermark using 13 queries for each token. We note that for Exp, the perplexity of the watermarked content is significantly higher than that of the unwatermarked content. This is mainly because Exp does not allow sampling in watermark embedding, which becomes a deterministic algorithm when the key is fixed. In contrast, our watermark removal attack generates content with much lower perplexity, making it comparable to unwatermarked content when the query number under different keys exceeds 13. This can be attributed to our attack functioning as a layer of random sampling. Unlike greedy sampling methods, we have a probability to sample the token with the highest unwatermarked probability (see Sec. 4, Appendix H, and Appendix I). The results of the three watermarks and two models prove that the watermark-removal attack exploiting the use of multiple watermark keys can effectively eliminate the watermarks while maintaining high output quality.

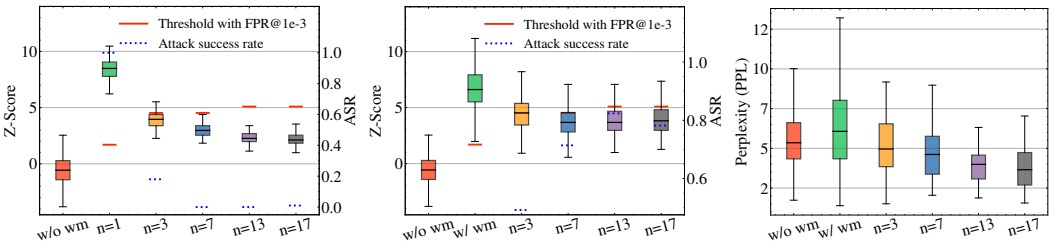

**(a)** Z-Score and attack success rate (ASR) of watermark stealing [25].

**(b)** Z-Score and attack success rate (ASR) of watermark-removal.

**(c)** Perplexity (PPL) of watermark-removal.

**Figure 11:** Spoofing attack based on watermark stealing [25] and watermark-removal attacks on Unigram watermark and LLAMA-2-7B model with different number of watermark keys $n$.

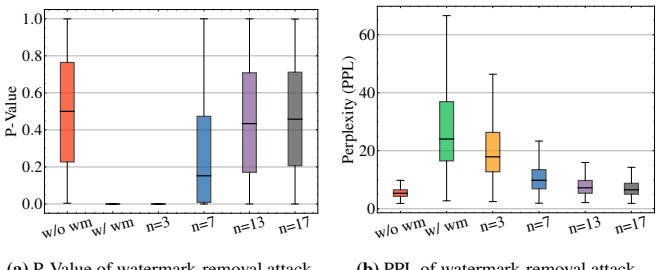

**(a)** P-Value of watermark-removal attack.   **(b)** PPL of watermark-removal attack.

**Figure 12:** Watermark-removal on Exp watermark [19] and LLAMA-2-7B model with multiple watermark keys.

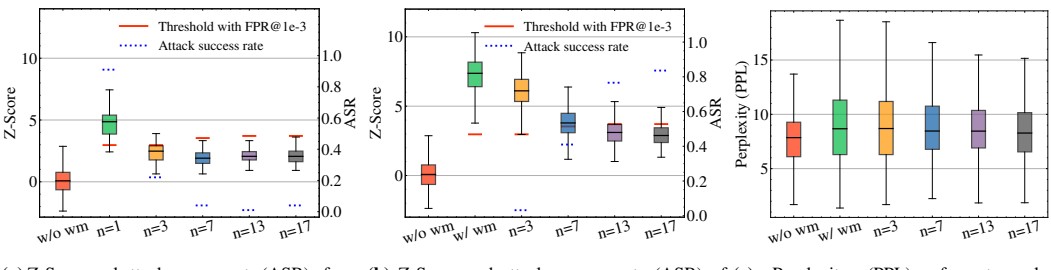

**(a)** Z-Score and attack success rate (ASR) of watermark stealing [25]. **(b)** Z-Score and attack success rate (ASR) of watermark-removal. **(c)** Perplexity (PPL) of watermark-removal.

**Figure 13:** Spoofing attack based on watermark stealing [25] and watermark-removal attacks on KGW watermark and OPT-1.3B model with different number of watermark keys $n$.

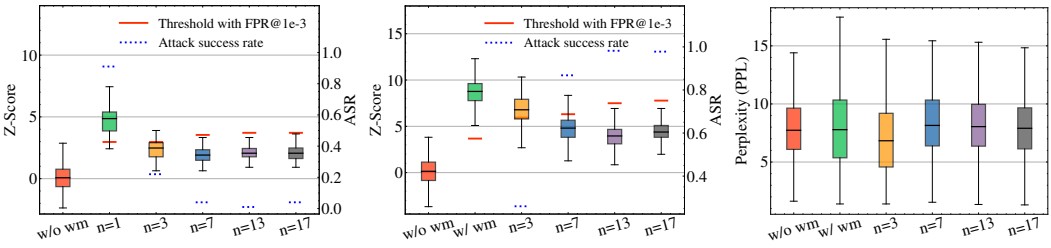

**(a)** Z-Score and attack success rate (ASR) of watermark stealing [25]. **(b)** Z-Score and attack success rate (ASR) of watermark-removal. **(c)** Perplexity (PPL) of watermark-removal.

**Figure 14:** Spoofing attack based on watermark stealing [25] and watermark-removal attacks on Unigram watermark and OPT-1.3B model with different number of watermark keys $n$.

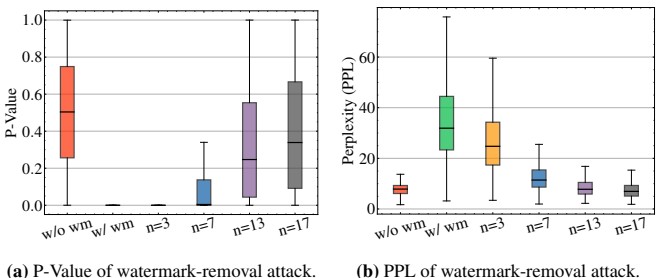

**(a)** P-Value of watermark-removal attack.   **(b)** PPL of watermark-removal attack.

**Figure 15:** Watermark-removal on Exp watermark [19] and OPT-1.3B model with multiple watermark keys.

## K Additional Results of Attacks Exploiting Detection APIs

We present the results of watermark-removal and spoofing attacks on OPT-1.3B model in Fig. 16 and Table 3. The results are consistent with the LLAMA-2-7B model presented in Sec. 6.1., with all the attack success rates higher than 75% using a small number of queries to the detection API

of around 3 per token. The results on OPT-1.3B model further demonstrate the effectiveness of our attacks exploiting the detection API.

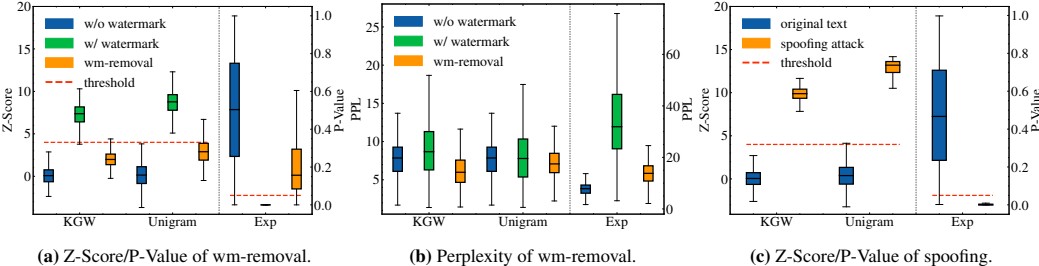

**(a)** Z-Score/P-Value of wm-removal.   **(b)** Perplexity of wm-removal.   **(c)** Z-Score/P-Value of spoofing.

**Figure 16:** Attacks exploiting detection APIs on OPT-1.3B model.

|  | wm-removal | | spoofing | |
|---|---|---|---|---|
|  | ASR | #queries | ASR | #queries |
| KGW | 0.99 | 2.87 | 1.00 | 2.96 |
| Unigram | 0.77 | 3.25 | 1.00 | 2.97 |
| Exp | 0.86 | 2.07 | 0.93 | 2.92 |

**Table 3:** The attack success rate (ASR), and the average query numbers per token for the watermark-removal and spoofing attacks exploiting the detection API on OPT-1.3B model.

## L  Additional Results of DP Defense

We present additional evaluation results of our defence technique that enhances the watermark detection by utilizing the techniques of differential privacy (see Sec. 6). Consistent with Sec. 6.3, we evaluate the utility of the DP defense as well as its performance in mitigating the spoofing attack exploiting the detection API. The results are shown in Fig. 17, Fig. 18, Fig. 19, Fig. 20, Fig. 21.

We first identify the optimal noise scale parameter $\sigma$ based on its detection accuracy and attack success rate, aiming for a drop in detection accuracy within $2\%$ and the lowest attack success rate. Then we assess the performance of the defense. Our findings across three watermarks and two models consistently demonstrate that we can significantly reduce the attack success rate to around or below $20\%$.

Our defense can be generalized to all LLM watermarking schemes. It allows us to substantially mitigate spoofing attacks exploiting the detection API while having negligible impact on utility.

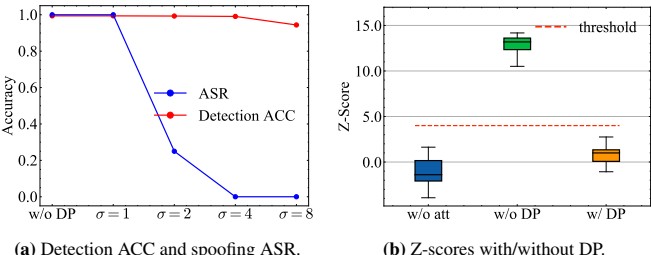

**(a)** Detection ACC and spoofing ASR.   **(b)** Z-scores with/without DP.

**Figure 17:** Evaluation of DP watermark detection on Unigram watermark and LLAMA-2-7B model. **(a).** Detection accuracy and spoofing attack success rate without and with DP watermark detection under different noise parameters. **(b).** Z-scores of original text without attack, spoofing attack without DP, and spoofing attacks with DP. We use the best $\sigma = 4$ from **(a)**.

## M  Additional Results of Larger Context Width and Using P-Values

In this section, we present more results of our attacks on larger context width $h = 4$ in the KGW watermark. The experiments are conducted on LLAMA-2-7B model. To demonstrate that changing

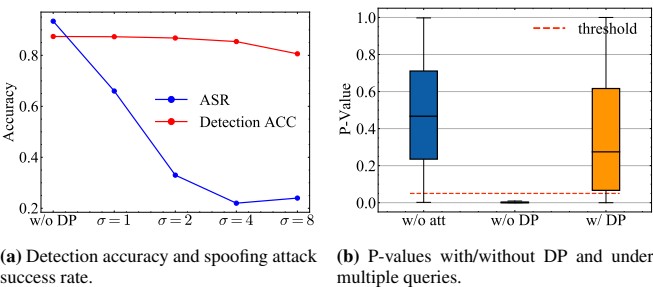

**(a)** Detection accuracy and spoofing attack success rate.

**(b)** P-values with/without DP and under multiple queries.

**Figure 18:** Evaluation of DP watermark detection on Exp watermark and LLAMA-2-7B model. **(a).** Detection accuracy and spoofing attack success rate without and with DP watermark detection under different noise parameters. **(b).** Z-scores of original text without attack, spoofing attack without DP, and spoofing attacks with DP. We use the best $\sigma = 4$ from **(a)**.

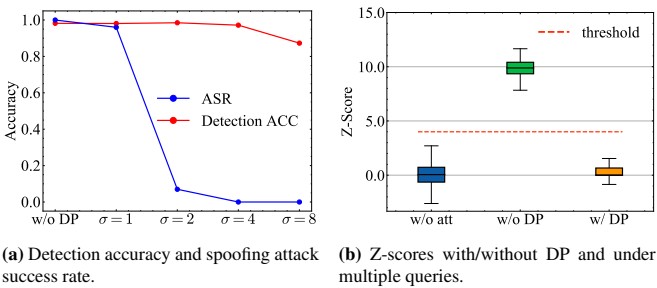

**(a)** Detection accuracy and spoofing attack success rate.

**(b)** Z-scores with/without DP and under multiple queries.

**Figure 19:** Evaluation of DP watermark detection on KGW watermark and OPT-1.3B model. **(a).** Detection accuracy and spoofing attack success rate without and with DP watermark detection under different noise parameters. **(b).** Z-scores of original text without attack, spoofing attack without DP, and spoofing attacks with DP. We use the best $\sigma = 4$ from **(a)**.

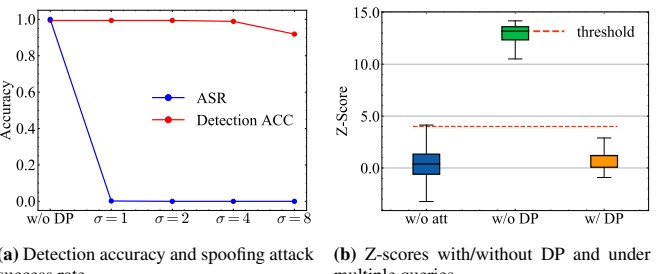

**(a)** Detection accuracy and spoofing attack success rate.

**(b)** Z-scores with/without DP and under multiple queries.

**Figure 20:** Evaluation of DP watermark detection on Unigram watermark and OPT-1.3B model. **(a).** Detection accuracy and spoofing attack success rate without and with DP watermark detection under different noise parameters. **(b).** Z-scores of original text without attack, spoofing attack without DP, and spoofing attacks with DP. We use the best $\sigma = 4$ from **(a)**.

to p-values has minor effects on the attack results, we adopt p-values here. We keep using z-scores in the main paper to ease the presentation (e.g., the p-values of the spoofing results will be too small to visualize). The results are shown in Fig. 22, Fig. 23, Fig. 24, We observe consistent results when using p-values in the KGW watermark and we expect minor influence on the results from this change since p-value is monotonic to z-score. We also get consistent observations for the scenarios of using larger context width $h$ in KGW watermark. Again, using larger $h$ enhances the resistance against watermark stealing but reduces robustness. Our experiments in Fig. 22, Fig. 23, Fig. 24 validate this. Fig. 22 shows that fewer edits are allowed for watermarked content with a larger $h$, indicating lower robustness. KGW recommends using $h < 5$ in their codebase for robustness, and no prior works we are aware of suggest using $h > 4$. Recent work [14] shows successful watermark stealing even with $h = 4$. Using multiple keys, as shown in Sec. 5 of our paper, mitigates stealing attacks, but introduces new attack vectors of watermark removal.

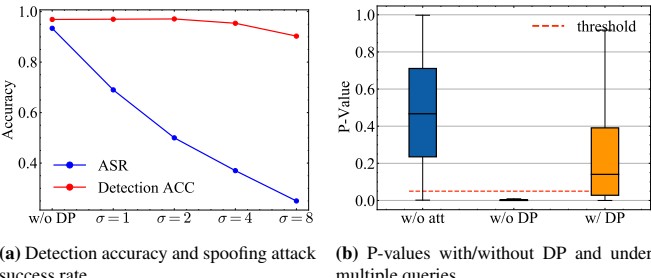

**(a)** Detection accuracy and spoofing attack success rate.

**(b)** P-values with/without DP and under multiple queries.

**Figure 21:** Evaluation of DP watermark detection on Exp watermark and OPT-1.3B model. **(a).** Detection accuracy and spoofing attack success rate without and with DP watermark detection under different noise parameters. **(b).** Z-scores of original text without attack, spoofing attack without DP, and spoofing attacks with DP. We use the best $\sigma = 4$ from **(a)**.

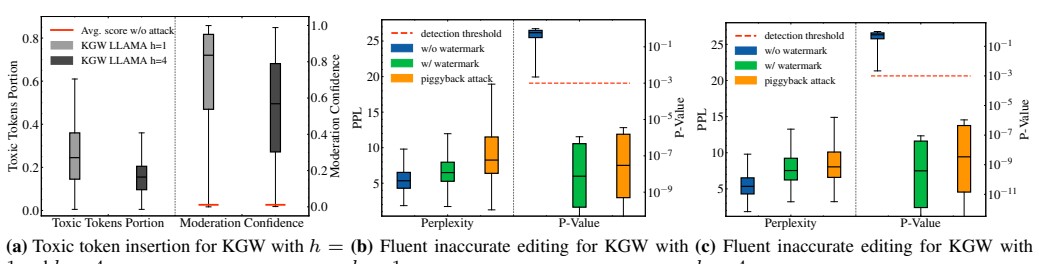

**(a)** Toxic token insertion for KGW with $h = 1$ and $h = 4$.

**(b)** Fluent inaccurate editing for KGW with $h = 1$.

**(c)** Fluent inaccurate editing for KGW with $h = 4$.

**Figure 22:** Piggyback spoofing of KGW watermark on LLAMA-2-7B model. We observe consistent results with Fig. 1. **(a)** The toxic token insertion works on both $h = 1$ and $h = 4$ with sumhash. KGW with $h = 4$ and sumhash is less robust, thus we can insert fewer toxic tokens. **(b, c)** The performances of fluent inaccurate editing for $h = 1$ and $h = 4$ are close. Changing z-statistics to p-value does not influence the attack performance.

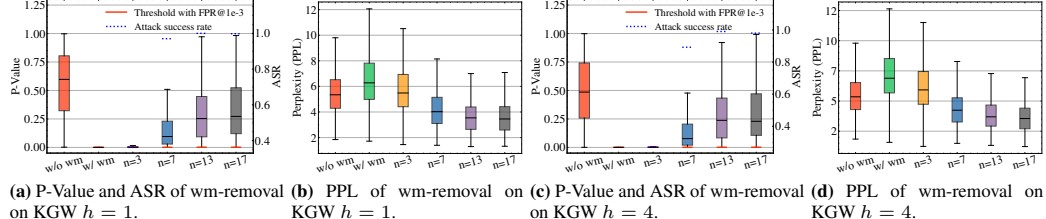

**(a)** P-Value and ASR of wm-removal on KGW $h = 1$.

**(b)** PPL of wm-removal on KGW $h = 1$.

**(c)** P-Value and ASR of wm-removal on KGW $h = 4$.

**(d)** PPL of wm-removal on KGW $h = 4$.

**Figure 23:** Watermark-removal attacks exploiting the use of multiple keys on KGW watermark ($h = 1$ and $h = 4$ with sumhash) and LLAMA-2-7B model with different numbers of watermark keys $n$. The attack success rates are based on the threshold with FPR@1e-3. We observe consistent results for different context widths $h$, and changing the detection metric from z-statistics to p-value does not influence the attack performance. The results are consistent with Fig. 2.

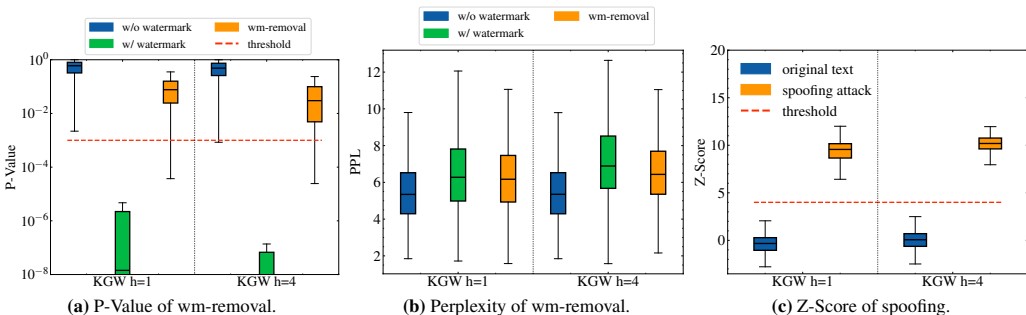

**(a)** P-Value of wm-removal.

**(b)** Perplexity of wm-removal.

**(c)** Z-Score of spoofing.

**Figure 24:** Attacks exploiting detection APIs on KGW watermark with different $h$, ($h = 1$ and $h = 4$ with sumhash) and LLAMA-2-7B model. Results are consistent with Fig. 3. We report the z-score for spoofing attacks, as the p-values of the spoofing results are too small to visualize. Changing z-score to p-value doesn't influence the attacks' performance and the results hold for different $h$.

