# OpenReview forum: "No Free Lunch in LLM Watermarking: Trade-offs in Watermarking Design Choices"
_NeurIPS.cc/2024/Conference — NeurIPS 2024 poster_

### Official Review · Reviewer_UAwD · 2024-06-17

**Soundness:** 3
**Presentation:** 3
**Contribution:** 2
**Rating:** 5
**Confidence:** 4

**Summary:**

This paper demonstrates that typical design choices in large language model (LLM) watermarking schemes result in significant trade-offs between robustness, utility, and usability. To navigate these challenges, this paper rigorously examines a series of straightforward yet effective attacks on prevalent LLM watermarking approaches and proposes practical guidelines and defenses to strengthen their security and practicality.

Specifically, the robustness of existing watermarking, a desirable property to mitigate removal attacks, can make the systems susceptible to piggyback spoofing attacks, which makes watermarked text toxic or inaccurate through small modifications. By proposing a novel attack, it is further shown that using multiple watermarking keys can make the system susceptible to watermark removal attacks. Finally, it is identified that public watermark detection APIs can be exploited by attackers to launch both watermark-removal and spoofing attacks. The paper proposes a defense strategy leveraging techniques from differential privacy to effectively counteract spoofing attempts.

**Strengths:**

The paper provides novel attack schemes for existing watermarking techniques and provides empirical evidence to support their claims, which help the community to better understand the tradeoff among the design of watermarking systems. In general, release detection API will always make the watermarking more vulnerable, and the proposed defense demonstrates an interesting connection to DP.

**Weaknesses:**

One of the biggest weaknesses of this paper is that the proposed attacks mainly explore the drawbacks of existing literature in [11,14,33], and some of the tradeoffs described in the paper are tied to these algorithms or specific formulations, which are not fundamental to the watermarking problem itself.

1.	The robustness issue discussed in Section 4 is mainly due to the specific definition of robustness Definition 3. As robustness is defined using editing distance, the piggyback spoofing attacks leverage the fact that we can significantly change the meaning of a sentence by editing very few tokens.
One way to address this issue is to define robustness using the semantics of the generated text instead of editing distance. If the meaning of the text remains similar, the watermarking should still be detectable; otherwise, the watermarking should disappear if the edits dramatically change the meaning of the text. Conceptually, I believe that there should not exist a tradeoff between robustness and spoofing attack. Empirically, the authors could conduct additional experiments on the following semantics-based watermarking. I am curious to see if the proposed attacks still work for semantic-based watermarking.

Liu, Aiwei, Leyi Pan, Xuming Hu, Shiao Meng, and Lijie Wen. "A semantic invariant robust watermark for large language models." ICLR 2024

Liu, Yepeng, and Yuheng Bu. "Adaptive Text Watermark for Large Language Models." ICML 2024

2.	As shown in "Adaptive Text Watermark for Large Language Models," by adding watermarking using semantics, the ASR of a spoofing attack is quite low without using multiple secret keys. In addition, it is likely that the same prompt will generate text with similar semantics, leading to a similar biased watermarking pattern. Therefore, it would be hard to do watermarking removal using the proposed attacks. The authors are encouraged to provide more discussion regarding the applicability of their empirical findings to different types of watermarks.

**Questions:**

I can imagine that similar issues discussed in the paper will also occur for watermarking images generated by the diffusion model. For example, the following two papers consider image watermarking removal using adversarial samples from the publicly available detection model.

Saberi, Mehrdad, Vinu Sankar Sadasivan, Keivan Rezaei, Aounon Kumar, Atoosa Chegini, Wenxiao Wang, and Soheil Feizi. "Robustness of ai-image detectors: Fundamental limits and practical attacks." arXiv preprint arXiv:2310.00076 (2023).

Jiang, Zhengyuan, Jinghuai Zhang, and Neil Zhenqiang Gong. "Evading watermark based detection of AI-generated content." In Proceedings of the 2023 ACM SIGSAC Conference on Computer and Communications Security, pp. 1168-1181. 2023.

This paper would greatly benefit from a thorough discussion of the differences and connections between these two problems by having a more thorough literature review.

**Limitations:**

Limitations are addressed in Appendix A of the paper.

---

> ### Author Rebuttal · Authors · 2024-08-07
>
> We thank the reviewer for their constructive comments. In the following, we respond to each question.
>
> ---
> >**Q1: One of the biggest weaknesses of this paper is that the proposed attacks mainly explore the drawbacks of existing literature in [11,14,33], and some of the tradeoffs described in the paper are tied to these algorithms or specific formulations, which are not fundamental to the watermarking problem itself.**
>
> **A1**: Please see global response **C2**.
>
> ---
> >**Q2: The robustness issue discussed in Section 4 is mainly due to the specific definition of robustness Definition 3. [...] I am curious to see if the proposed attacks still work for semantic-based watermarking.**
>
> **A2**: Thanks for this suggestion; please see global response **C2.1**.
>
> ---
> >**Q3: As shown in "Adaptive Text Watermark for Large Language Models," [...] The authors are encouraged to provide more discussion regarding the applicability of their empirical findings to different types of watermarks.**
>
> **A3**: Thanks for the comments. Our findings in the tradeoffs of using multiple watermark keys do not apply to the semantics-based watermarks, as they rely on a separate semantics embedding model instead of determining the watermark logit bias using a watermark key. The general spoofing attack considered in [2] works for the PRF-based robust watermarks, and we may need substantial modifications to the general spoofing attacks to gain a better attack performance in semantics-based watermarks. For instance, instead of estimating the watermarked tokens’ distributions by providing uniformly distributed prompts, the attacker will need to carefully construct the prompts to guarantee that the outputs will be semantically close, such that they can gain some information on which portion of the tokens is more likely to appear in a specific semantic context by making a large number of queries. To defend against such potential attacks, one option could be for the model provider to introduce randomness into the semantics embedding model. For example, multiple random semantics embedding models could be used during inference, similar to the setting of using multiple watermark keys, to make it more resistant to watermark stealing. In this case, our findings will still apply, but this would need a more thorough and rigorous investigation.
>
> As we mentioned in the global response **C2**, attacking semantics-based watermarks is not the focus of our paper, but we will clarify this and provide corresponding discussions in the revision.
>
> [2] Liu et al. Adaptive Text Watermark for Large Language Models. ICML 24
>
> ---
> >**Q4: I can imagine that similar issues discussed in the paper will also occur for watermarking images generated by the diffusion model. [...] This paper would greatly benefit from a thorough discussion of the differences and connections between these two problems by having a more thorough literature review.**
>
> **A4**: We agree that the attacks utilizing the detection API can be generalized to the image watermarks, as the attackers can adopt a similar oracle attack pipeline. The attackers will need to integrate domain-specific constraints to guarantee that the generated sentences or images are meaningful and high-quality. We will discuss potential opportunities and challenges in extending our attacks to image watermarks in the limitations/future work section of our revision (e.g., using the 1 extra camera-ready page if necessary). Thanks for this suggestion.

---

### Official Review · Reviewer_Hb7x · 2024-06-19

**Soundness:** 3
**Presentation:** 2
**Contribution:** 1
**Rating:** 5
**Confidence:** 5

**Summary:**

The paper details three different attacks on LLM watermarking, targeting watermark removal and spoofing:
A1: spoofing by taking advantage of the robustness of the watermark
A2: removal by taking advantage of multiple watermarking keys
A3: removal by taking advantage of a public detector

Attacks are followed by some guidelines/defenses.

**Strengths:**

Attack A2 is very original (AFAIK). The proposed defense against A3 based on DP is interesting.

**Weaknesses:**

W1. Relevance
I have been a watermarking practitioner for years. I have never seen the following proposals:
- Robust watermarking as proof of content authenticity (A1-Section 4)
A robust watermarking detector distinguishes 2 hypotheses:
H_0: content is not watermarked (by this technique and that secret key)
H_1: content has been watermarked **and** potentially modified.
So, I do not perceive A1 (Section 4) as an attack but more as a misunderstanding or a misuse of robust watermarking by the authors.
I strongly disagree with Guideline #1 (line 206), which suggests lowering the robustness of robust watermarking. This hardly makes sense.
The recommendation should be to combine two schemes: robust watermarking and fragile (digital-signature-based) watermarking.
Robust watermarking for authenticity is wrong. Fragile watermarking for IA-gen detection is wrong as well.

- Watermarking detectors should be public (A3-Section 6)
This leads to oracle attacks well documented in the watermarking literature of the 200s (nowadays called black-box attacks). The authors study even the easiest case where the attacker observes a soft decision (Z statistics): he can observe if any single modification lowers the detection score. A harder and more relevant case would be a Yes/No decision. But I am not even recommending that setup for security reasons. In short, the novelty (the attack against a specific LLM watermarking) is narrow. "*It is still an open question whether watermark detection APIs should be made publicly available*". It is not an open question, and the conclusion of this section is absolutely not a surprise.

W2. State-of-the-art
The experimental setup considers 3 schemes with default parameters. Here default parameters mean parameters as appearing in the very first version of these papers. Yet, since then, we know that these choices were not adequate.
- Use of Z-statistics. They have been shown to be suboptimal and inaccurate, leading to theoretical FPRs that are way off the empirical FPRs. I recommend using p-values (empirically validated). See "Three Bricks to Consolidate Watermarks for LLMs" from P. Fernandez.
- The choice 'h=1' in KGW is not recommended (Needless to say that Unigram --where 'h=0'-- is even worse). Again, it leads to inaccurate FPR. Moreover, it is not secure (watermarking stealing attack). The question is whether your attack still holds with a larger h (with a proper hash function, not those implemented based on min of hashes).

W3. Key management
As far as A2 is concerned, I think the key issue is key management. The literature offers 2 flavors: randomly picking a key in a small key space (Kuditipudi) or using a hash of the previous tokens (Aaronson, Kirchenbauer). Section 5 only investigates the first option.

**Questions:**

Q1. Attack A3. Section 6. The assumption is that the watermarked LLM provides the top-5 tokens. Although some LLM APIs like OpenAI provide this information, I have some doubts that a **watermarked** LLM would do this. Especially, what are the top-5 tokens for Exp? Are they computed before or after the Gumbel trick?

Q2. I am a bit surprised that $\sigma = 4$ doesn't impact the detection performance.
Without DP: I suppose that w/o watermark $Z\sim\mathcal{N}(0;1)$ and with watermark $Z\sim\mathcal{N}(\mu;1)$, with $\mu \approx 6$ (according to Fig. 3.a). With a threshold set to 4, this makes P_FP ~ 3e-5,  P_TP ~ 0.98, for an accuracy of 0.99.

With DP: The variances are now equal to $\sqrt{1+\sigma^2}$. This makes P_FP ~ 0.17, P_TP ~ 0.69, for an accuracy of 0.76.... Quite far away from what you get. More importantly, these two cases should be compared for a fixed P_FP so that the threshold with DP should be higher.  Reporting only the accuracy is masking the fact that P_FP is way higher.

**Limitations:**

The title and introduction discuss LLM watermarking in general, but the paper is based on three particular schemes. I agree that these three are the most well-known. It is questionable whether other more exotic schemes (like semantic-based) are also vulnerable.

One may easily think of easy counter-attacks: forbidding querying the LLM with the same prompt repeatedly (or the same prompt plus one extra token), forbidding querying the detector with too similar text, etc.

---

> ### Author Rebuttal · Authors · 2024-08-07
>
> We appreciate the reviewer's constructive comments. Please refer to **C1** in the global response for our clarifications on the positioning and contributions of our work. Our work studies the feasibility and ramifications of potential attacks, with the goal of better informing the public and the LLM watermarking community about key design tradeoffs. Below, we respond to each question.
>
> ---
> >**Q1: Robust watermarks are not suitable for proof of content authenticity.**
>
> **A1**: We agree robust watermarks are more suited for AI content detection, while fragile (signature-based) watermarks are used for authentication.
>
> Our contribution in this section is a rigorous study of the tradeoff between robustness and spoofing resistance, therefore showing that no single scheme is sufficient to protect against both types of attacks. Our experiments show the extent to which robust watermarks are vulnerable. We’ve recommended using fragile watermarks for spoofing defense in Sec 4.2 (L 215), but it is also known that fragile watermarks such as signatures are not robust to editing.
>
> We emphasize that increasing a watermark’s robustness to editing diminishes its suitability for content authenticity. This tradeoff is valuable information for the LLM watermark community. Indeed, we note that we are not the first to explore spoofing and robust watermarks, as recent works [1,2] have proposed more complex spoofing attacks on a specific set of robust watermarks. In contrast to these works, we explore a more simple and general piggyback spoofing attack that allows us to explore the inherent trade-off between spoofing and robustness. The potential for spoofing has also been cited as a major barrier to industrial LLM watermarking deployment [3], making it important to study this tradeoff rigorously.
>
> We agree that the current guideline can be revised to better convey our conclusions. We will follow your suggestion to revise our guideline to: `Guideline #1: Robust watermarks are vulnerable to spoofing attacks and are not suitable as proof of content authenticity alone. To mitigate spoofing while preserving robustness, it may be necessary to combine additional measures such as signature-based fragile watermarks.`
>
> ---
> >**Q2: The findings of attacks exploiting public detection API are not surprising.**
>
> **A2**: In Sec 6 (L 288), we stated that “Although this (public detection API) makes it easier to detect watermarked text, it is commonly acknowledged that it will make the system vulnerable to attacks. Here, we study this statement more precisely by examining the specific risk trade-offs that exist, as well as introducing a novel defense that may make the public detection API more feasible in practice.”
>
> We mentioned the use of public detection APIs is an ‘open question’ given recent activity in the community—for example, recent keynotes mention such settings [4], and commercial AI content detection services that return confidence scores to users [5]. However, we can remove this sentence to avoid misunderstandings. Despite the risks, public detection APIs have a lot of benefits, such as improving transparency in AI usage and supporting regulatory compliance. We believe providing a detection API isn't a simple yes or no question, and that there are different knobs one can tune when providing such an API. Our work both provides and explores such knobs, so that the risks/benefits of public detection APIs may be considered for practical deployment.
>
> Oracle attacks have existed for decades, but rigorously exploring them in the context of LLM watermarks is necessary. We show that making detection scores differentially private can effectively mitigate the spoofing attack without compromising detection accuracy (see **A7**). Our findings can help to enable public detection APIs deployment, inspiring future LLM watermark designs.
>
> ---
> >**Q3: I recommend using p-values as a detection metric.**
>
> **A3**: Thanks for this suggestion; please see global response **C6**.
>
> ---
> >**Q4: Whether your attack still holds with a larger h?**
>
> **A4**: See global response **C3**.
>
> ---
> >**Q5: Key management.**
>
> **A5**: See global response **C4**.
>
> ---
> >**Q6: The assumption of watermarked LLM providing the top-5 tokens.**
>
> **A6**: See global response **C5**.
>
> ---
> >**Q7: DP noise scale.**
>
> **A7**: For KGW and Unigram, we add noise to the z-scores. Sensitivity varies with sentence length (e.g., $\Delta=\frac{h+1}{\sqrt{\gamma(1-\gamma)l}}$ for replacement editing, where $l$ is the sentence length, $h,\gamma$ are watermark parameters). The actual noise scale is proportional to $\sigma\Delta$. For a 200-token sentence, $h=1,\gamma = 0.5,\sigma = 4$, the noise scale is 0.8. We’ve tested FPR with DP defense on OpenGen dataset, and FPRs are close to 0 (below 1e-3) using our recommended noise scale. We’ll explain more clearly in the text to avoid confusion.
>
> ---
> >**Q8: Are other more exotic schemes (like semantic-based) also vulnerable?**
>
> **A8**: See global response **C2**.
>
> ---
> >**Q9: One may easily think of easy counter-attacks: forbidding querying the LLM with the same prompt repeatedly (or the same prompt plus one extra token), forbidding querying the detector with too similar text, etc.**
>
> **A9**: In Guidelines #2 and #3, we recommended “defense-in-depth” techniques like anomaly detection, query rate limiting, and user verification. However, with just rate limiting, our attacks remain possible, as service providers can’t always ensure trusted users. Thus, it’s important to consider the tradeoffs when deploying LLM watermarking systems.
>
> ---
> [1] Jovanović et al. Watermark Stealing in Large Language Models. ICML 24
>
> [2] Sadasivan et al. Can AI-generated text be reliably detected? arXiv 23
>
> [3] Somesh Jha. Keynote at SaTML 2024-Watermarking (The State of the Union). 2024
>
> [4] Scott Aaronson. Watermarking of large language models. 2023
>
> [5] AI Purity; GPTZero; Winston AI

---

> > ### Comment · Reviewer_Hb7x · 2024-08-09
> >
> > **About Attack A1**
> >
> > I prefer this new guideline. A LOT.
> >
> > **About Attack A2**
> > - There is a small contradiction in the text. The attack is motivated by Eq. (4) known as distortion-freeness or unbiasedness. Yet, the experiment considers KGW or Unigram, which are not distortion-free or unbiased. This is understandable because Watermarking Stealing only holds for green-list-based methods (AFAIK). It might be good to warn the reader.
> > - The results also deeply rely on the way detection proceeds with multiple keys. There are plenty of variants. It amounts at computing a p-value per key, then aggregating these p-values into one statistic and computing the associated *global* p-value. Here the aggregation is the min operator over p-values (i.e., max operator over the score). I believe more robust alternatives are Fisher, Edgington, or Harmonic Mean aggregations. Anyway, I just mean that the results deeply rely on the setup and some precautions in the text are welcome.
> > https://en.wikipedia.org/wiki/Harmonic_mean_p-value
> > - Another guideline could be: Stay away from Watermark Stealing and Never use multiple keys.
> >    + Either prefer *NON* green-list-based methods like EXP. This is backed by Fig. 12 & 15.
> >    + Either use a green-list-based method with a proper cryptographic hash function (not makeshifts like HashMin or HashSum which are flawed) and a large h.
> >
> > BTW, about EXP, I don't understand line 729
> > >  the use of a large number of watermark keys is inherent in their design, which defaults to 256.
> >
> >
> > **About Attack A3**
> >
> > > Oracle attacks have existed for decades...
> >
> > Why don't you cite them?
> >
> > > “defense-in-depth” techniques such as anomaly detection
> >
> > Would you mind providing references, please.
> >
> > > We show that making detection scores differentially private...
> >
> > About DP,  why does the sensitivity depend on $h$?

---

> > > ### Author Response · Authors · 2024-08-10
> > >
> > > Thanks for the reviewer’s timely reply. We respond to follow-up comments below:
> > >
> > > >**Attack A1.**
> > >
> > > We will update this guideline in our revision. Thanks again for the reviewer’s suggestion.
> > >
> > > ---
> > >
> > > >**There is a small contradiction in the text.**
> > >
> > > In Sec 5 (L 230), we mentioned that Exp is rigorously unbiased (the $\epsilon$ in Eq. 4 is negligible), and KGW and Unigram slightly shift the watermarked distributions (the $\epsilon$ in Eq. 4 could be large and won’t converge with the increasing of key numbers). We will emphasize this point and also clarify that watermark stealing does not work on the rigorously unbiased watermarks in the revision.
> > >
> > > ---
> > >
> > > >**The results also deeply rely on the way detection proceeds with multiple keys.**
> > >
> > > Our watermark-removal attack exploiting the use of multiple keys is not dependent on the aggregation method as we do not rely on the server’s watermark detection in this attack. However, the tradeoff analysis and the sweet spot for the number of the keys may slightly change given the different detection performance for various aggregations. We will add a paragraph to discuss this interesting problem in the revision; thanks for bringing it up.
> > >
> > > ---
> > >
> > > >**Another guideline could be: Stay away from Watermark Stealing and Never use multiple keys.**
> > >
> > > We want to clarify that the original Exp watermarking scheme inherently uses multiple keys in their setup: it maintains a predefined set of watermark keys, and at each time of model inference, it will randomly sample a key (a starting key index) from the pool, as also mentioned in the reviewer’s previous review (randomly picking a key in a small key space (Kuditipudi)). The number of keys is defaulted to 256 in their paper’s evaluation and codebase.
> > >
> > > Since the use of multiple keys is inherent in Exp, it can defend against watermark stealing attacks at the cost of being vulnerable to our watermark-removal attacks. The results in Figs. 12 & 15 show that we can effectively remove the watermark in Exp when n=7, given that the p-value of this attack is significantly large. However, for n=3, our watermark-removal attack does not work. To defend against watermark-removal, Exp needs to consider using fewer keys or limit query rates for users. We note that using a smaller number of keys like 3 would destroy the distortion-free guarantee and make Exp vulnerable to watermark stealing.
> > >
> > > We will follow your suggestion to recommend the use of larger h with proper hash functions in the guideline, and point out its tradeoff between robustness in the revision, as we have discussed in the global response **C4**.
> > >
> > > ---
> > >
> > > >**Citations for oracle attacks and “defense-in-depth” techniques.**
> > >
> > > We are happy to provide citations in our revision to support these points. Specifically, for oracle attacks, there are related works in both cryptography [1,2] and watermark analysis [3,4]. For defense-in-depth techniques we will cite [5,6].
> > >
> > > [1] Bleichenbacher, Daniel. Chosen ciphertext attacks against protocols based on the RSA encryption standard PKCS# 1. CRYPTO 1998.
> > >
> > > [2] Cramer, Ronald, and Victor Shoup. Design and analysis of practical public-key encryption schemes secure against adaptive chosen ciphertext attack. SIAM Journal on Computing 2003.
> > >
> > > [3] Linnartz, Jean-Paul MG, and Marten Van Dijk. Analysis of the sensitivity attack against electronic watermarks in images. Information Hiding. Springer, 1998.
> > >
> > > [4] Kalker, Ton, J-P. Linnartz, and Marten van Dijk. Watermark estimation through detector analysis. Proceedings 1998 International Conference on Image Processing. 1998.
> > >
> > > [5] Bau, Jason, et al. State of the art: Automated black-box web application vulnerability testing. IEEE S&P 2010.
> > >
> > > [6] Sommer, Robin, and Vern Paxson. Outside the closed world: On using machine learning for network intrusion detection. IEEE S&P 2010.
> > >
> > > ---
> > >
> > > >**About DP, why does the sensitivity depend on h?**
> > >
> > > In KGW, considering replacement editing, each edit will change the hash that is used to split the green and red token lists for the length of context width tokens, which is h. This will affect at most $h+1$ tokens (including the token being edited) in terms of whether they are detected in the green or red list. Thus, the z-score sensitivity is bounded by $\frac{h+1}{\sqrt{\gamma(1-\gamma)l}}$.
> > >
> > > ---
> > >
> > > Please let us know if you have further comments, questions, or suggestions. We thank the reviewer again for their constructive feedback. If you believe that some of your key concerns have been addressed, we would greatly appreciate it if you are willing to revisit your score.

---

> > > > ### Comment · Reviewer_Hb7x · 2024-08-10
> > > >
> > > > > We want to clarify that the original Exp watermarking scheme inherently uses multiple keys in their setup
> > > >
> > > > Yes! my fault, sorry. I though that EXP was referring to the original Aaronson scheme. I see now that it refers to the version of Kuditipudi.
> > > >
> > > > > Citations for oracle attacks and “defense-in-depth” techniques.
> > > >
> > > > I don't see why [1,2,4,5]  are relevant here. You better stick to watermarking citations. For defense I would recommend:
> > > >
> > > > Mauro Barni, Pedro Comesaña-Alfaro, Fernando Pérez-González, and Benedetta Tondi "Are you threatening me?: Towards smart detectors in watermarking", Proc. SPIE 9028, Media Watermarking, Security, and Forensics 2014, 902806 (19 February 2014); https://doi.org/10.1117/12.2036415
> > > >
> > > > El Choubassi, Maha, and Pierre Moulin. "On reliability and security of randomized detectors against sensitivity analysis attacks." IEEE Transactions on Information Forensics and Security 4.3 (2009): 273-283.
> > > >
> > > > The last ref shows that the idea of adding noise to the sufficient statistics computed by the watermark detector is rather old.

---

> > > > > ### Author Response · Authors · 2024-08-11
> > > > >
> > > > > Thanks for your follow-up comments. In our previous response, we provided references for classic oracle attacks in cryptography [1,2], oracle attacks in the context of watermarking [3,4], and vulnerabilities and anomaly detection techniques in web applications that can be potentially generalized to detection APIs [5,6]. We are happy to cite the works you mentioned.
> > > > >
> > > > > The last paper you mentioned in particular introduces a randomized detector designed for DCT image watermarks. The protocol works for watermarks where the watermarked content $Y$ is a linear mapping of the original data vector $X$ and the watermark pattern vector $W$: $Y = X + W$, with $X_i$ being iid. The DCT coefficients of images can be treated as iid, but in LLM watermark detectors, the input samples (which are input tokens) can be highly correlated. The different watermark embedding pipeline and non-iid input tokens make it nontrivial to apply this method to LLM watermarks. We will mention this interesting work in our revision as an early work exploring randomization in watermark detection schemes.

---

### Official Review · Reviewer_iwzS · 2024-07-13

**Soundness:** 3
**Presentation:** 2
**Contribution:** 3
**Rating:** 6
**Confidence:** 3

**Summary:**

In this work, the authors reveal new attack vectors including watermark-removal attacks and spoofing attacks that exploit common features and design choices of LLM watermarks. Besides, the authors propose a defense utilizing the ideas of differential privacy, which increases the difficulty of spoofing attacks.

**Strengths:**

1. The research question in this paper is interesting and is a hot topic in the field of LLM watermark.
2. The paper is well-written.
3. The experimental data and results presented in this paper are extensive.

**Weaknesses:**

1. In Sec 3.1, what are the differences between "piggyback" and "general" spoofing attacks? Specifically, what does "piggyback" refer to?

2. In Sec 3.1, regarding attacks on detection APIs, the reviewer is confused by the statement "the attacker can auto-regressively synthesize (toxic) sentences." What does this mean?

3. Regarding attacks discussed in Sec. 5, the attackers can discover watermarking rules by observing a large amount of watermarked text, thus enabling attacks. The original KGW paper in Sec. 5 mentions using a large context width h to defend against such attacks. However, this paper lacks explanation and discussion regarding the parameter h.

4. In the detection API attack, the description of the adversary's capabilities is unclear. In Sec 6.1, the authors assume the adversary can access the target watermarked LLM's API and query watermark detection results. This implies the adversary can generate watermarked text and obtain detection results for any given text. But why can the adversary generate a list of possible replacements for x_i? Does this mean the adversary can access the perturbed probability distribution and logits of tokens? If so, this seems to exceed the stated capabilities of the adversary.

5. There are some minor issues, such as the undefined notation "V^{\ast}" in the definition in Sec 3.

**Questions:**

Please answer all points in the Weaknesses section.

**Limitations:**

The authors have adequately addressed the limitations.

---

> ### Author Rebuttal · Authors · 2024-08-07
>
> We thank the reviewer for their constructive comments. In the following, we respond to each question.
>
> ---
> >**Q1: In Sec 3.1, what are the differences between "piggyback" and "general" spoofing attacks? Specifically, what does "piggyback" refer to?**
>
> **A1**: Piggybacking (Sec 4, L 142) is a classic attack in computer networks, where the attacker tags along with another person who is authorized to gain entry into a restricted area. The general spoofing attacks for LLM watermarks [1, 2] usually require the attacker to first estimate the watermark pattern by making a large number of queries (observe millions of watermarked tokens) to the watermarked LLM, and then they can create malicious content with a target watermark embedded but in fact it is not generated by the watermarked LLM to ruin the reputation of LLMs.
>
> Our piggyback spoofing attack does not require estimating the watermark pattern, instead, we launch the attack based on the content generated by the watermarked LLM, which is similar to piggybacking in computer networks where the attacker relies on well-established authorization. The attacker’s goal of piggyback spoofing is the same as the general spoofing attack as they both aim to create malicious/inaccurate content with a target watermark embedded. However, a benefit of our attack is that it has much weaker assumptions on the attacker’s ability. The attacker can simply exploit the robustness property of the watermarks and maliciously edit the watermarked content without altering the watermark detection result to generate malicious but watermarked content to ruin the LLM’s reputation.
>
> [1] Jovanović et al. Watermark Stealing in Large Language Models. ICML 24
>
> [2] Sadasivan et al. Can AI-generated text be reliably detected? arXiv 23
>
> ---
> >**Q2: In Sec 3.1, regarding attacks on detection APIs, the reviewer is confused by the statement "the attacker can auto-regressively synthesize (toxic) sentences." What does this mean?**
>
> **A2**: In the attacks exploiting the detection APIs, the attacker will generate sentences auto-regressively, similar to how LLMs generate sentences. That is, the attacker will select each token based on the prior tokens and the detection results. Please also refer to Alg.1 and Alg.2 in the Appendix J of our paper. We will clearly explain this in the revision.
>
> ---
> >**Q3: Regarding attacks discussed in Sec 5, the attackers can discover watermarking rules by observing a large amount of watermarked text, thus enabling attacks. The original KGW paper in Sec 5 mentions using a large context width h to defend against such attacks. However, this paper lacks explanation and discussion regarding the parameter h.**
>
> **A3**: Please see global response **C4**.
>
> ---
> >**Q4: In the detection API attack, the description of the adversary's capabilities is unclear. In Sec 6.1, the authors assume the adversary can access the target watermarked LLM's API and query watermark detection results. This implies the adversary can generate watermarked text and obtain detection results for any given text. But why can the adversary generate a list of possible replacements for x_i? Does this mean the adversary can access the perturbed probability distribution and logits of tokens? If so, this seems to exceed the stated capabilities of the adversary.**
>
> **A4**: Please see global response **C5**.
>
> ---
> >**Q5: There are some minor issues, such as the undefined notation $V^{\ast}$ in the definition in Sec 3.**
>
> **A5**: The $V^{\ast}$ refers to a sequence of tokens, where each token belongs to the vocabulary set $V$. We will clearly explain this in the revision.

---

> > ### Comment · Reviewer_iwzS · 2024-08-12
> >
> > The reviewer thanks the authors for the response. The score is kept the same.

---

### Official Review · Reviewer_YBsi · 2024-07-13

**Soundness:** 4
**Presentation:** 3
**Contribution:** 3
**Rating:** 7
**Confidence:** 4

**Summary:**

This paper explores the vulnerabilities and trade-offs in watermarking schemes for large language models (LLMs). It highlights how common design choices in these schemes, aimed at ensuring robustness, multiple key usage, and public detection, make them susceptible to simple yet effective attacks. The authors demonstrate that robust watermarks, intended to prevent removal, can be easily exploited through piggyback spoofing attacks that insert toxic or inaccurate content while maintaining the watermark. Additionally, using multiple watermark keys to defend against watermark stealing inadvertently increases vulnerability to watermark removal attacks. Public detection APIs, while useful for verifying watermarked content, are shown to be exploitable for both removal and spoofing attacks. Through empirical evaluations on state-of-the-art watermarks (KGW, Unigram, Exp) and models (LLAMA-2-7B, OPT-1.3B), the study rigorously demonstrates these vulnerabilities and the resulting trade-offs between robustness, utility, and usability. The paper proposes potential defenses, including the use of differential privacy techniques in detection APIs, and offers guidelines for designing more secure watermarking systems. Ultimately, the study underscores the importance of carefully considering watermarking design choices to balance security and utility, calling for further research to develop robust defenses and optimize these trade-offs.

**Strengths:**

+ The paper rigorously explores various common watermarking design choices and demonstrates their susceptibility to simple yet effective attacks. It highlights the fundamental trade-offs between robustness, utility, and usability, which are crucial for understanding the limitations of current watermarking methods.

+ The paper provides an insightful discussion of the inherent trade-offs in watermarking design, such as the balance between watermark robustness and vulnerability to spoofing attacks. This helps in understanding the complexities involved in creating effective watermarking.

+ The authors propose potential defenses and guidelines to enhance the security of LLM watermarking systems. These recommendations are valuable for practitioners looking to deploy more secure watermarking solutions in practice.

**Weaknesses:**

+ The citation in Line 248 and Figure 2 is not correct. The authors are supposed to cite [1].

+ Lack of discussions in the Publicly-Detectable Watermarking [2], which compromises robustness to defend against spoofing attacks.




[1] Nikola Jovanovic, Robin Staab, and Martin Vechev. Watermark stealing in large language models. arXiv preprint arXiv:2402.19361, 2024.
[2] Fairoze, Jaiden, et al. "Publicly detectable watermarking for language models." arXiv preprint arXiv:2310.18491 (2023).

**Questions:**

+ How generalizable are the findings? Would the vulnerabilities and trade-offs identified apply to all types of LLMs, or are they specific to certain architectures or applications?
+ What are the limitations of the attack methods presented in this study? Are there scenarios where these attacks might not be effective?
+ Are there any practical considerations or potential drawbacks to implementing DP defense mechanisms in real-world systems?

**Limitations:**

See the weaknesses.

---

> ### Author Rebuttal · Authors · 2024-08-07
>
> We thank the reviewer for their constructive comments. In the following, we respond to each question.
>
> ---
> >**Q1: The citation in Line 248 and Figure 2 is not correct. The authors are supposed to cite [1].**
>
> **A1**: Thanks for pointing this out. We will fix this typo in the revision to avoid confusions.
>
> ---
> >**Q2: Lack of discussions in the Publicly-Detectable Watermarking [2], which compromises robustness to defend against spoofing attacks.**
>
> **A2**: There exist some recent works [1,2] that study mitigating the spoofing attack vulnerabilities in robust watermarks. The high-level idea is to embed a cryptographic signature into the subsequent tokens, and the signatures are computed using the first $m$ high-entropy tokens and the secret key. They further incorporate error correction code to make the design robust.
>
> As also mentioned by the reviewer, such designs are not as robust as the watermarks we study as they prioritize the resistance against spoofing instead of strong robustness. For instance, by simply modifying the first $m$ tokens, the signature check no longer passes. The designs of these works are consistent with our findings in the piggyback spoofing attacks: to defend against spoofing attacks, the design needs to incorporate less robust (or even non-robust) signature-based watermarks. We will include these related works in the revision to provide a more comprehensive study.
>
> [1] Zhou et al. Bileve: Securing Text Provenance in Large Language Models Against Spoofing with Bi-level Signature. arXiv 24
>
> [2] Fairoze, et al. Publicly detectable watermarking for language models. arXiv 23
>
> ---
> >**Q3: How generalizable are the findings? Would the vulnerabilities and trade-offs identified apply to all types of LLMs, or are they specific to certain architectures or applications?**
>
> **A3**: Please see global response **C2**.
>
> ---
> >**Q4: What are the limitations of the attack methods presented in this study? Are there scenarios where these attacks might not be effective?**
>
> **A4**: As we have discussed in **C2**, the semantics-based watermarks rely on a high-quality semantics embedding model instead of using secret keys to embed the watermark. As such designs fundamentally differ from the watermarks we study, our findings of using multiple watermark keys are not applicable here. We will include a limitation and future work section in the revision to discuss this issue and present a more comprehensive study.
>
> ---
> >**Q5: Are there any practical considerations or potential drawbacks to implementing DP defense mechanisms in real-world systems?**
>
> **A5**: DP adds noise to the watermark detection results. The service provider needs to determine the optimal noise scale, as larger noise will make the detection inaccurate, and less noise will be ineffective to defend against attackers. According to our empirical findings, we can find a sweet point to achieve both high detection accuracy (low FPR) and low attack success rate. Overall, we believe that our DP defense can potentially make the detection API publicly available while protecting the secret watermark pattern information without sacrificing detection accuracy.

---

> > ### Comment · Reviewer_YBsi · 2024-08-12
> >
> > Thank you for the author's response. My concerns have been thoroughly addressed, and I recommend incorporating the discussions into the revision. This paper has the potential to significantly impact the field of LLM watermarks. I will maintain my score and advocate for its acceptance.

---

### Author Rebuttal · Authors · 2024-08-07

We appreciate all reviewers’ constructive comments. Below we clarify our contributions, respond to common questions, and present new experimental results.

>**C1: Clarification on the contributions and positioning of our work.**

Our work explores attacks that exploit design choices of common LLM watermarks. While these design choices may enhance robustness, resistance against watermark stealing attacks, and public detection ease, we show that they also allow malicious actors to launch attacks that can easily remove the watermark or damage the model's reputation. Although some of our high-level take-aways may confirm common beliefs (e.g., the risk of spoofing robust watermarks as noted by Reviewer Hb7x), we disagree with the implication that the feasibility and ultimate ramifications of such attacks are thus unworthy of scientific study—particularly given that these design choices have been adopted/explored both in recent research and in practical deployment. Further, our work questions common folklore (such as the inability to use public detection APIs), showing that attacks on these systems may be mitigated with our novel DP-inspired defense. Overall, our goal is to rigorously study the risks and benefits of LLM watermark design tradeoffs, and to distill these results into a set of take-aways that can better inform the public and LLM watermarking community. We consider these take-aways particularly important as the research community grows and the use of LLM watermarking systems increases, potentially out of the hands of a select set of domain experts.

---
>**C2: Generalizability of our attacks. (Hb7x, UAwD, YBsi)**

We focus on three SOTA PRF-based robust watermarks, which are a natural set to explore given their popularity and formal cryptographic guarantees. There are other promising watermarks like the semantics-based watermarks as the reviewers mentioned. While attacking semantics-based watermarks is outside the scope of our study, we agree with Reviewer UAwD that this is an interesting direction to explore, and have provided an initial exploration below. We will discuss generalizing our attacks to other watermarks including semantics-based watermarks as a potential avenue of future work in our revision.

>**C2.1: Piggyback spoofing on semantics-based watermarks. (UAwD)**

Semantics-based watermarks use embedding models to capture sentence semantics and bias LLM predictions. Robustness ensures semantically close sentences yield similar watermark patterns. We agree spoofing attacks are harder if we assume perfect semantics embedding models. However, inaccuracy in the embedding can make spoofing possible.

As a proof of concept, we attacked the SIR watermark [1], and present a concrete example in Tab.1 of the submitted PDF to show piggyback spoofing is possible. We deem this an interesting future direction to rigorously explore and will add discussions in the revision.

[1]Liu et al. A semantic invariant robust watermark for large language models. ICLR 24

---
>**C3: Consistent attack performance for larger h. (Hb7x)**

Our results hold for any h and hash function in KGW watermark. Increasing h makes brute-force watermark stealing harder, but our attacks don’t depend on h or hash functions. With the latest KGW codebase, we use h=4 and sumhash in new experiments, observing consistent results with h=1 for all attacks, as shown in Figs.1-3 in the submitted PDF.

---
>**C4: Discussions on the tradeoff of context width h. (Hb7x, iwzS)**

We primarily explored the fundamental tradeoffs in using multiple watermark keys, which prior works have underexplored. Tradeoffs in context widths (h) are discussed in prior works [1-3]. Using larger h enhances the resistance against watermark stealing but reduces robustness. Our new experiments validate this. Fig.1 shows that fewer edits are allowed for watermarked content with a larger h, indicating lower robustness. KGW recommends using h<5 in their codebase for robustness, and no prior works we are aware of suggest using h>4. Recent work [1] shows successful watermark stealing even with h=4. Using multiple keys, as shown in Sec 5 of our paper, mitigates stealing attacks, but introduces new attack vectors of watermark removal. We will add a discussion on larger h in the revision.

[1]Jovanović et al. Watermark Stealing in Large Language Models. ICML 24

[2]Kirchenbauer et al. A watermark for large language models. ICML 23

[3]Zhao et al. Provable Robust Watermarking for AI-Generated Text. ICLR 24

---
>**C5: Clarifications on obtaining top-5 tokens from the watermarked LLM. (Hb7x, iwzS)**

In our watermark-removal attack with detection APIs, we assume the attacker can generate a short list of replacements for the current token. We used the setting of returning top-5 tokens by the watermarked LLM API because it’s beneficial to users and is currently used in commercial non-watermarked LLM services including OpenAI [1]. For instance, it can help understand model confidence, enable debugging, make custom sampling strategies, etc. One of the goals of our paper is to point out how existing LLM deployment practices can lead to attacks if watermarking is integrated. The fact that this API is vulnerable to our attacks illustrates our point. We will clearly state the attacker's assumptions in the revision.

[1]OpenAI API. https://platform.openai.com/docs/api-reference/completions/create

---
>**C6: Using p-value instead of z-score as the detection metric. (Hb7x)**

We will follow the reviewer’s suggestion to change the detection metric from z-score to p-value for KGW and Unigram. P-values are used for Exp in our paper, and the observations are consistent with KGW and Unigram. We expect no impact on results from this change since p-value is monotonic to z-score. Figs.1-3 in the submitted PDF also show consistent attack performance using p-values for KGW, and we will add results for Unigram in the revision.

---

### Decision · Program_Chairs · 2024-09-25

**Decision:**

Accept (poster)

**Comment:**

This LLM watermarking paper has received broadly supportive reviews that either firmly or partially recommend acceptance.  The reviewers appreciated value in the study of trade-offs between design choices in LLM watermarking for attack mitigation.  The more supportive reviews considered the work rigorous and with the potential to significantly impact the practice of LLM watermark design.  Other reviewer comments were broadly positive and prompted an exchange to clarify details of the baselines and experimental setup, which are either confirmed or appear to have been satisfied by the authors’ rebutall and responses.  The authors should be diligent to deliver the clarifications detailed in the discussion in the final version.  Overall the AC supports the acceptance of the paper – the reviews are positive and the work appears to be timely and potentially impactful in the practice of LLM watermarking.